# Mof-associated complexes have overlapping and unique roles in regulating pluripotency in embryonic stem cells and during differentiation

Sarina Ravens[1], Marjorie Fournier[1], Tao Ye[2], Matthieu Stierle[1], Doulaye Dembele[2], Virginie Chavant[3], Làszlò Tora[1]*

[1]Cellular signaling and nuclear dynamics program, Institut de Génétique et de Biologie Moléculaire et Cellulaire (IGBMC), CNRS UMR 7104 - Inserm U 964, Université de Strasbourg, Illkirch, France; [2]Microarrays and deep sequencing platform, Institut de Génétique et de Biologie Moléculaire et Cellulaire (IGBMC), CNRS UMR 7104 - Inserm U 964, Université de Strasbourg, Illkirch, France; [3]Proteomics platform, Institut de Génétique et de Biologie Moléculaire et Cellulaire (IGBMC), CNRS UMR 7104 - Inserm U 964, Université de Strasbourg, Illkirch, France

*For correspondence: laszlo@igbmc.fr

Competing interests: The authors declare that no competing interests exist.

**Abstract** The histone acetyltransferase (HAT) Mof is essential for mouse embryonic stem cell (mESC) pluripotency and early development. Mof is the enzymatic subunit of two different HAT complexes, MSL and NSL. The individual contribution of MSL and NSL to transcription regulation in mESCs is not well understood. Our genome-wide analysis show that i) MSL and NSL bind to specific and common sets of expressed genes, ii) NSL binds exclusively at promoters, iii) while MSL binds in gene bodies. Nsl1 regulates proliferation and cellular homeostasis of mESCs. MSL is the main HAT acetylating H4K16 in mESCs, is enriched at many mESC-specific and bivalent genes. MSL is important to keep a subset of bivalent genes silent in mESCs, while developmental genes require MSL for expression during differentiation. Thus, NSL and MSL HAT complexes differentially regulate specific sets of expressed genes in mESCs and during differentiation.

## Introduction

Pluripotent mouse embryonic stem cells (mESCs) have the ability to self-renew or to differentiate into all cell types. Specific transcription factors like Oct4, Sox2, and Nanog form a core transcriptional network, which is required for the maintenance of mESC pluripotency (*Orkin et al., 2008*). Chromatin-modifying enzymes further regulate transcriptional mESCs networks and cellular differentiation processes and can be associated with activation or repression of genes (*Orkin and Hochedlinger, 2011*). Histone acetylation is important for mESC pluripotency and is regulated by the concerted action of histone acetyltransferases (HATs) and histone deacetylases (HDACs) (*Meshorer and Misteli, 2006*). Acetylation of histone proteins leads to an open and dynamic chromatin conformation allowing an active transcription state, which is also a signature of mESC pluripotency (*Meshorer, 2007*; *Niwa, 2007*; *Efroni et al., 2008*). During differentiation of mESCs, the overall transcription rates decrease, whereas the chromatin structure becomes more compact with a global reduction of histone H3 and H4 acetylation. In line with the requirement of histone acetylation in mESC maintenance and differentiation, genetic deletion or knockdown of several HATs affects mESC pluripotency (*Lin et al., 2007*; *Fazzio et al., 2008*; *Gupta et al., 2008*; *Thomas et al., 2008*; *Zhong and Jin, 2009*; *Li et al., 2012*).

**eLife digest** Embryonic stem cells are special cells that have the ability to become many different types of cells, such as skin, muscle, or neuronal cells. This process is called differentiation. They can also undergo a process called self-renewal to produce more embryonic stem cells. These processes are controlled by a complex network of enzymes, and the production of these enzymes depends on various genes within the organism being expressed as proteins.

The DNA that holds the genetic information inside cells spends most of its time wrapped around proteins called histones: this allows the DNA molecules—which can be up to several metres long in some species—to fit inside the cell nucleus; it also protects the DNA molecules, which are quite fragile, from damage. Enzymes that attach chemical groups called acetyl groups to histones have a central role in controlling the self-renewal and differentiation of embryonic stem cells.

Mof is an enzyme that attaches an acetyl group to a specific position in a particular histone. It is a subunit within two larger protein complexes that were originally identified in flies: the male-specific lethal (MSL) complex, which is only found in male flies, and the non-specific lethal (NSL) complex, which is found in both male and female flies. These complexes have been widely studied in flies, and the role of the Mof enzyme is also reasonably well understood in mammals. However, the roles of the MSL and NSL protein complexes in mammals are not fully understood.

Ravens et al. have now used a combination of a technique called ChIP-seq (which can identify binding sites anywhere in the genome) and genetic 'knock down' experiments to explore the roles of these two complexes in mouse embryonic stem cells and neuronal progenitor cells.

There is some overlap between the genes that the complexes act on. However, NSL acts on some genes than MSL does not act on, and vice versa. NSL mostly acts on genes that have 'housekeeping' functions and are expressed in many different cell types. MSL binds more to genes that are specific to embryonic stem cells, and acts on genes required for the development of neuronal progenitor cells. This means that NSL regulates the growth of embryonic stem cells, whereas MSL controls their development and differentiation.

HATs can be classified into two predominant families: the GCN5-related *N*-acetyltransferase (GNAT) family (i.e. Gcn5 and p300) and the Moz-Ybf2/Sas3-Sas2-Tip60 (MYST) family (i.e., Tip60 and Mof [male absent on the first]) (reviewed in *Kimura et al., 2005*). These enzymes often function as part of multi-protein co-activator complexes (reviewed in *Lee and Workman, 2007*). Mof (also known as Kat8 or Myst1), is a MYST-type HAT specific for histone H4 lysine 16 acetylation (H4K16ac) (*Hilfiker et al., 1997*; *Smith et al., 2001*, *2005*; *Taipale et al., 2005*) and has been shown to be the catalytic subunit of two distinct protein complexes in *Drosophila* (d) and mammals: the male-specific lethal (MSL) and the non-specific lethal (NSL) complexes (*Smith et al., 2005*; *Mendjan et al., 2006*; *Cai et al., 2010*; *Raja et al., 2010*). In *Drosophila*, the dMSL complex is targeted to transcribed regions of male X-chromosomal genes, where it mediates dosage compensation (reviewed in *Straub and Becker, 2007*; *Gelbart and Kuroda, 2009*; *Conrad and Akhtar, 2011*). In contrast, dNSL is present at gene promoters of male and female chromosomes, where it regulates transcription of housekeeping genes (*Prestel et al., 2010*; *Raja et al., 2010*; *Feller et al., 2012*; *Lam et al., 2012*). Mof itself and subunits of the Mof-containing dMSL and dNSL HAT complexes are required for the binding of the two *Drosophila* Mof-containing complexes at promoters and gene bodies, which leads to H4K16 acetylation and gene expression (*Raja et al., 2010*; *Kadlec et al., 2011*).

*Inactivation of Mof* in mice (m) leads to early embryonic lethality as *Mof*⁻/⁻ embryos fail to develop beyond the expanded blastocyst stage and die at implantation (*Gupta et al., 2008*; *Thomas et al., 2008*). Mof deletion correlated with cell cycle defects and cell death. Moreover, mESCs could not be derived from *Mof*⁻/⁻ mouse embryos. In agreement, it was shown that Mof plays an essential role in the maintenance of mESC pluripotency (*Li et al., 2012*). H4K16 acetylation levels were undetectable in *Mof*⁻/⁻ embryos, whereas the acetylation of other histone lysine residues was unaffected (*Thomas et al., 2008*). Surprisingly, loss of H4K16 acetylation upon neuronal differentiation of mESCs did not alter higher-order chromatin compaction (*Taylor et al., 2013*). Moreover H4K16ac and Mof were reported to be present at the transcription start sites (TSSs) of expressed genes in mESCs (*Li et al., 2012*; *Taylor et al., 2013*).

Details on the function of the mammalian Mof-containing MSL and NSL complexes have only recently started to emerge and revealed that Mof fulfils different functions within the two HAT complexes. Human (h) MSL complex is composed of the subunits MSL1, MSL2, MSL3, and MOF (*Smith et al., 2005*; *Mendjan et al., 2006*), while the hNSL complex is composed of nine subunits: NSL1, NSL2, NSL3, MCRS1, WDR5, PHF20, HCF1, OGT1, and MOF (*Mendjan et al., 2006*; *Cai et al., 2010*). Mammalian NSL complex appears to have broader substrate specificity than the MSL complex, as it is also able to acetylate non-histone targets (*Li et al., 2009*). However, the function of MSL and NSL complexes in mammalian cells and especially their role in establishing mESC pluripotency in not well understood.

To better understand the role of MSL and NSL in gene regulation and their individual contribution in epigenetic changes in mESCs, we have analysed these two Mof-containing complexes by chromatin immunoprecipitation coupled with high throughput sequencing (ChIP-seq) and by shRNA knockdown (KD) experiments in mESCs. The obtained genome-wide binding maps show that MSL and NSL locate to a large number of expressed genes and each complex has a distinct binding profile at promoters or gene bodies. Our combined ChIP-seq and KD data indicate that MSL and NSL have a combinatorial effect on a given set of genes, whereas some specific loci are only MSL- or only NSL-dependent. Our data indicate that NSL binds exclusively at promoters, while MSL binds more in gene bodies. We show that in mESCs NSL regulates cell growth whereas MSL is the main HAT complex acetylating histone H4K16. MSL is present at mESC-specific genes. Moreover, MSL binds to and regulates developmental genes in mESCs and during differentiation. Altogether our data demonstrate that MSL and NSL complexes are present at expressed genes in mESCs, but that MSL is essential for regulation of key mESC-specific and bivalent developmental genes.

## Results

### Both Msl1 or Nsl1 incorporate in their respective Mof-containing HAT complexes in mESCs

To understand the global role of the two Mof-containing complexes in chromatin remodelling and how this regulates genes linked to self-renewal, proliferation, and/or differentiation, we set out to analyse the genome-wide binding of MSL and NSL in mESCs. To this end, we raised antibodies targeting Msl1 or Nsl1, which are specific subunits of the MSL or NSL complexes, respectively, and are known to play a role in the assembly and the regulation of these complexes (*Raja et al., 2010*; *Kadlec et al., 2011*). The specificity of the purified antibodies was demonstrated by western blot assays and immunoprecipitations followed by mass spectrometry using the multidimensional protein identification technology (MudPIT) (*Figure 1*). Western blot assays indicated that both of the generated antibodies are specific (*Figure 1A,B*). In addition, both antibodies immunoprecipitated (IP-ed) the endogenous MSL and NSL complexes with the previously described polypeptide composition (*Cai et al., 2010*; *Figure 1C*). Importantly, Mof was identified in both IP-ed MSL or NSL complexes, in the same range of abundance than Msl1, or Nsl1 (*Figure 1C*, *Figure 1—source data 1* for all identified proteins by MudPIT). Gel filtration followed by western blot analyses further indicated that Msl1 and Nsl1 are only present in Mof-containing complexes as they have eluted from the Superose 6 column in the same molecular weight containing fractions as their respective entire endogenous MSL (about 240 kDa), or NSL (about 800 kDa) complexes (*Figure 1D*). Of note the enzymatic subunit Mof was detected in the respective MSL and NSL complexes, but in addition as a potentially free form in the 50 kDa range fractions (*Figure 1D*). These results together demonstrate the incorporation of all nuclear Msl1, or Nsl1, together with Mof, in their respective endogenous complexes and a fraction of 'free' Mof that is not present in either MSL or NSL.

### MSL binds mainly to gene bodies, while NSL to promoter regions in mESCs

To characterize the genome-wide role of MSL and NSL complexes, we carried out ChIP-seq analysis in mESCs using the above-characterized anti-Msl1 and anti-Nsl1 antibodies. The obtained binding maps of Msl1 and Nsl1 in mESCs were then compared at the UCSC genome browser to publicly available ChIP-seq data for Mof, H4K16 acetylation (H4K16ac), RNA Polymerase II (Pol II), and DNAse hypersensitive sites (DHS). At a representative genomic locus, Nsl1 peaks were detected at the TSSs of four expressed genes, where they co-localized with Mof, Pol II and DHSs, whereas Msl1 binding peaks were

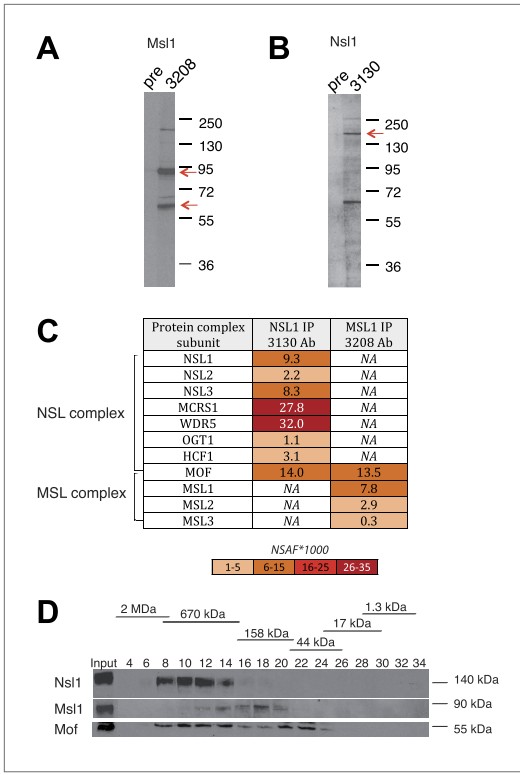

**Figure 1**. Msl1 and Nsl1 incorporate into endogenous complexes in mESCs. (**A** and **B**) Western Blot analysis using the raised anti-Msl1 (3208) or anti-Nsl1 (3130) antibodies on nuclear extracts. Preimmune sera (pre) were used as negative controls. (**C**) Anti-Msl1 (3208) or anti-Nsl1 (3130) antibodies were used to immunoprecipitate protein complexes from mESC nuclear extracts. The IP-ed complexes were then analysed by multidimensional protein identification technology (MudPIT). The identified MSL- or NSL-containing complex proteins and their relative protein abundance in the samples are represented by normalized spectral abundance factor (NSAF) (*Zybailov et al., 2006*). NSAF allows the comparison of abundance of individual proteins in multiple independent samples and in multiprotein complexes (*Florens et al., 2006*; *Paoletti et al., 2006*). The colour intensity reflects of the NSAF values multiplied by 1000 (as indicated). (**D**) Gel filtration of mESC nuclear extracts. Every second fraction eluted from a Superose 6 column was analysed for the presence of Nsl1, Msl1, and Mof by Western Blot. Molecular weight markers for the corresponding fractions are indicated on the top of the panel.

The following source data are available for figure 1:

**Source data 1**. List of identified proteins of MudPIT analyses.

usually broader and together with H4K16ac downstream of Pol II peaks (*Figure 2A*). As previously reported, Mof is present at promoters, gene bodies (GBs) and intergenic regions (*Li et al., 2012*).

Using MACS14 algorithm (*Zhang et al., 2008*) we determined high-confidence binding sites (peaks) for Msl1 or Nsl1 (*Figure 2—figure supplement 1A*, *Figure 2—source data 1*) and selected peaks with various tag densities for ChIP-qPCR validation. The Msl1 and Nsl1 enrichments at five different loci as detected by ChIP-qPCR faithfully reflected the tag densities measured by ChIP-seq (*Figure 2—figure supplement 1B,C*). To further verify the specificity of the Msl1 and Nsl1 ChIP-seq results, we used lentiviral small hairpin (sh) RNA vectors to knockdown (KD) Msl1 or Nsl1 in mESCs (*Figure 2—figure supplement 2A–D*) and tested by ChIP-qPCR the decrease of Msl1 or Nsl1 binding at the TSSs and in the GBs of two genes that were co-bound by these factors (*Figure 2—figure supplement 2E,D*). The predominant binding of Msl1 to GBs was lost upon Msl1 KD, whereas Nsl1 binding to TSS was reduced following Nsl1 depletion. These results confirm our ChIP-seq analyses.

Next, we asked whether the two complexes bind to common or different loci. A pairwise comparison of the MSL or NSL enrichment at all high confidence binding loci revealed that the binding of both complexes show two populations and have a Pearson correlation coefficient of 0.23 (p-value=$1.88 \times 10^{-160}$) (*Figure 2B*). This indicates a significant overlap between Msl1 and Nsl1 binding populations, but suggests also a differential genome-wide binding of MSL and NSL. To know at which genomic regions the identified peaks localize, each peak was annotated either to promoter, GB (containing introns, exons, untranslated regions and transcription termination sites together) or intergenic regions. 74% of all Msl1 peaks are detected at GBs (*Figure 2C*). In contrast, the majority of identified Nsl1 peaks are present at promoter regions (67%) (*Figure 2C*). Moreover, only about 10% of all Msl1- or Nsl1-binding sites map to intergenic regions (as defined above, excluding introns). The majority of the 9890 Msl1-, or 6251 Nsl1-specific binding sites are at promoter and/or GB regions (*Figure 2C*), and after removal of redundant genes, we defined 5844 Msl1- and 4755 Nsl1-bound genes (*Figure 2—figure supplement 1A*, *Figure 2—source data 2*). As only 10% of the binding sites were detected at intergenic regions, we focused our further analyses on the role of MSL and NSL complexes in gene regulation at the promoter and/or GB regions.

To understand the genome-wide binding of MSL and NSL, we compared by k-means clustering either Msl1, or Nsl1 binding profiles with that of Mof and the presence of H4K16ac at 30,300 ENSEMBL

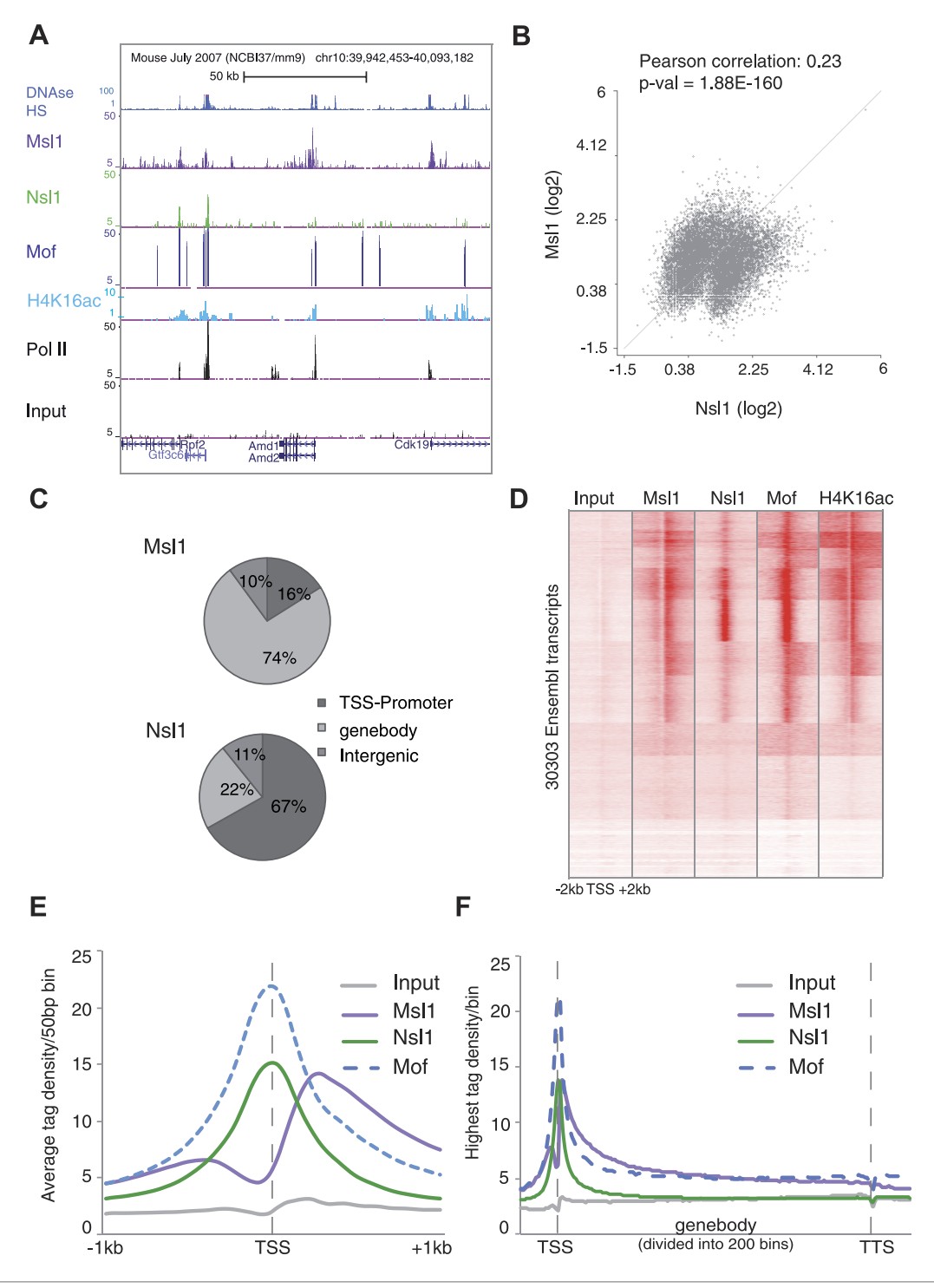

**Figure 2**. Distinct binding profiles of Msl1 and Nsl1 at active genes. (**A**) UCSC genome browser tracks representing Msl1, Nsl1, H4K16ac, Mof and Pol II ChIP-seq data. The Input serves as control. (**B**) Scatter Plot showing the Pearson correlation between Msl1 and Nsl1 densities at all identified MACS14 peak regions. Densities were normalized to the control (Input) and represented as log2 values. (**C**) Mapping of Msl1 and Nsl1 identified MACS14 peaks to different genomic regions (promoter-TSS, genebody [GB] or intergenic regions) using HOMER (*Heinz et al., 2010*). Identified peaks are listed in *Figure 2—figure supplement 1A* and *Figure 2—source data 1*. Validation of identified peaks is shown in *Figure 2—figure supplement 1B,C*. (**D**) Heatmap showing k-means clustering of Msl1, Nsl1, Mof and H4K16ac using the TSSs of all ENSEMBL transcript IDs as reference coordinates. Densities are

*Figure 2. Continued on next page*

*Figure 2. Continued*

presented ±2 kb around reference coordinates. Input serves as negative control. (**E** and **F**) Average binding profiles of Msl1, Nsl1 and Mof (**E**) at a region of +1 kb around the annotated TSSs and (**F**) 1 kb upstream of the TSS, in the GB and 1 kb downstream of the TTS. Only Nsl1 or Msl1 positive genes were taken into consideration. The Input serves as control and tag densities were normalized to the input. See *Figure 2—figure supplement 2* for validation of ChIP-seq data.

The following source data and figure supplements are available for figure 2:

**Source data 1**. List of Msl1 and Nsl1 MACS14 peaks.

**Source data 2**. List of Msl1 and Nsl1 positive genes.

**Figure supplement 1**. Identification and validation of Msl1 and Nsl1 binding sites.

**Figure supplement 2**. Knockdown (KD) of Msl1 or Nsl1 through lentiviral shRNA vectors.

---

transcription start sites (TSSs). In good agreement with our results showing that in mESCs Msl1 and Nsl1 incorporate in the endogenous MSL or NSL complexes, respectively, genome-wide Mof binding overlaps with that of Msl1 and Nsl1 around most of the TSSs, which are also H4K16ac positive (*Figure 2D*).

Interestingly, at promoters, MSL and NSL complexes have distinct binding profiles. Nsl1 and Mof show a sharp binding peak centred at the TSSs, while the average Msl1 binding profile is similar to H4K16ac (see below) and extends downstream from the TSSs in the GB regions (*Figure 2E*). Moreover, the Msl1 and Mof signals are enriched downstream of promoters at GBs, whereas the control and the Nsl1 signals are not (*Figure 2F*). Altogether our results demonstrate that MSL and NSL bind mostly to distinct sites in mESCs. NSL binds directly to the TSS region of genes, while the genome-wide location of MSL is both at TSSs and downstream of the TSSs of bound genes.

## MSL and NSL bind to active genes, but are differently related to gene expression

To assess the relationship between Msl1 and Nsl1 binding and gene expression in mESCs, we took advantage of available RNA-seq data (*Tippmann et al., 2012*) and compared the average expression of Msl1-, or Nsl1-bound genes (in median log2 FPKM values) to that of all ENSEMBL genes (*Figure 3A*). The median expression values for Msl1-, or Nsl1-positive genes were significantly higher as compared to all ENSEMBL genes, demonstrating that Msl1 and Nsl1 are mostly present at expressed genes in mESCs.

To determine whether the binding strength of Msl1 or Nsl1 correlates with gene expression, we compared Msl1 and Nsl1 enrichment around TSSs with gene expression data from the corresponding bound genes. Msl1- or Nsl1-positive genes were divided into five categories according to their expression levels (*Figure 3B–D*). As a control, in the same five categories densities of Pol II peaks at promoters correlated with gene expression with decreasing Pol II densities from highly to poorly expressed genes (*Figure 3A*; *Barski et al., 2007*). Importantly, the boxplot representation revealed a similar correlation as Pol II between Msl1 binding and gene expression, indicating that the stronger the gene is expressed the higher Msl1 and Pol II are enriched at the binding sites (*Figure 3B,C*). In contrast, there is no significant difference of the Nsl1 median values between the five groups, indicating that Nsl1-binding to promoters is not proportional with the level of expression (*Figure 3D*). Our results thus demonstrate that both Msl1 and Nsl1 bind to active genes, but that only the binding strength of Msl1, and not that of Nsl1, correlates with mRNA levels, suggesting a different dynamic and/or functional behaviour of the two complexes at the regulated loci.

## The localization of MSL, but not that of NSL, overlaps with the presence of H4K16ac

As H4K16 is a known target of Mof in *Drosophila* and mammals (*Hilfiker et al., 1997*; *Smith et al., 2001, 2005*; *Taipale et al., 2005*), we compared Msl1 or Nsl1 binding sites with the presence of H4K16ac. Our scatter plot analyses indicate that there is a general overlap of Msl1 or Nsl1 with

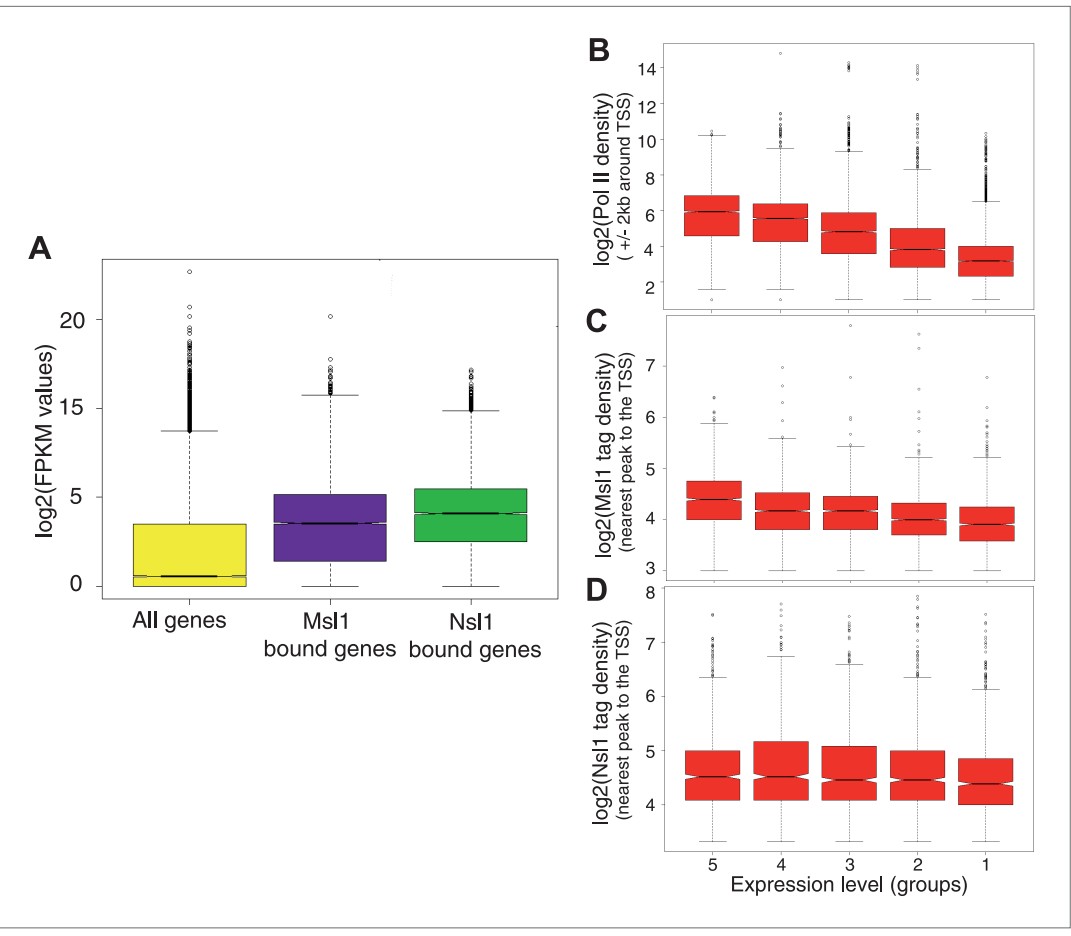

**Figure 3**. Msl1 and Nsl1 bind to active genes, but participate differentially to gene expression. (**A**) Boxplots showing the log2 of RNA FPKM expression values from mESCs of all analysed, Msl1-, or Nsl1-bound ENSEMBL genes. (B, C and **D**) RNA expression values are ranked into five groups, where group 5 represents the highest RNA expression level and group 1 the lowest (see bottom of panels **B**–**D**). Boxplots show the tag density of the nearest peak to the TSS for (**B**) Pol II, (**C**) Msl1 and (**D**) Nsl1 tag densities around the TSSs at the five groups. Only density values higher than zero were taken into consideration. The median is different between groups, if the notches of the boxplots do not overlap.

H4K16ac, whereas the correlation between Msl1 binding sites and H4K16ac is better (Pearson correlation coefficient 0.57) than between Nsl1 and H4K16ac (Pearson correlation coefficient 0.32), which is also reflected in the corresponding p-values (*Figure 4A,B*). The comparison of the distribution patterns of Msl1, Nsl1, Pol II and H4K16ac around the TSSs (±2 kb) of all Msl1- and Nsl1-bound genes further indicates that the Msl1 binding profile is more similar to the genome-wide presence of H4K16ac, than that of Nsl1 (*Figure 4C*). H4K16ac levels are enriched downstream of the TSSs overlapping with the binding profile of Msl1 (*Figure 4C*). In contrast, the centre of the Nsl1 binding profile centred at the TSS region does not overlap with that of the H4K16ac peak (*Figure 4C*). These binding profiles suggest a link between H4K16 acetylation and the MSL HAT complex (*Figure 4*).

## MSL is the main H4K16 acetylase in mESCs

Although it was demonstrated that Mof depletion in embryos results in a loss of H4K16ac (*Gupta et al., 2008*; *Thomas et al., 2008*), the exact contribution of the two Mof-containing HAT complexes to H4K16 acetylation remains to be determined. To address this question, we analysed the global acetylation of H4K16 after Msl1 or Nsl1 KD and also quantified acetylation of H4K5 and H4K8, two other proposed substrates for hNSL in differentiated human cells (*Cai et al., 2010*). Western blot analyses of total histone proteins from mESCs expressing shRNAs targeting Msl1 or Nsl1 revealed a

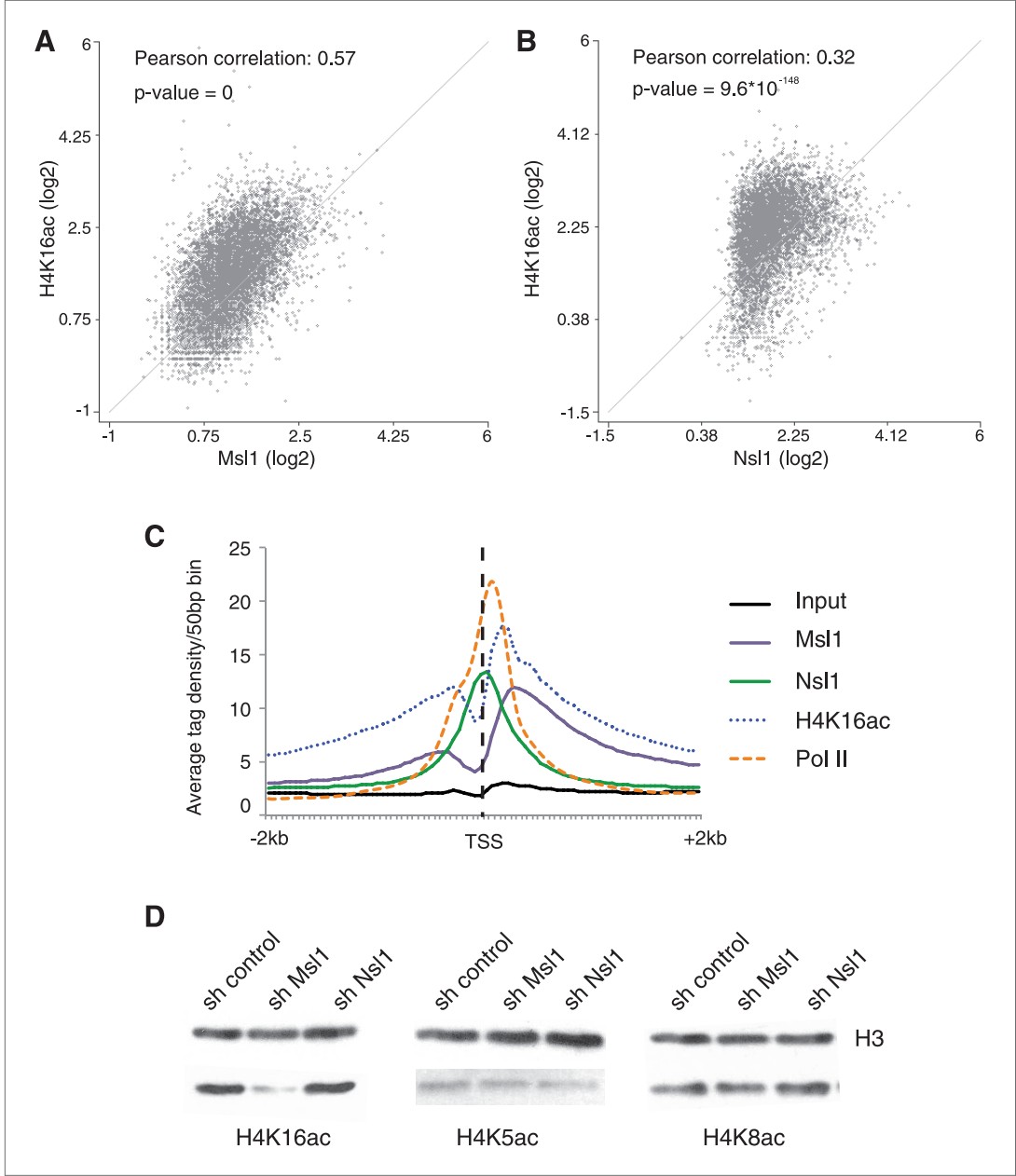

**Figure 4**. MSL affects H4K16 acetylation in mESCs. (**A** and **B**) Scatter Plots indicating the Pearson correlation and Pearson p-values between H4K16ac and Msl1 (**A**) or Nsl1 (**B**) densities at Msl1 peaks or Nsl1 peaks. Log2 represented tag densities were calculated at peak regions and normalized to the control (Input) data set. (**C**) Average binding profiles of Msl1, Nsl1, Pol II and H4K16ac at a region of +2 kb around all ENSEMBL promoters. Only Nsl1 and Msl1 positive genes are taken into consideration. The input serves as control and tag densities are normalized to the input. (**D**) mESCs were treated for 5 days with lentiviral vectors expressing sh control, sh Msl1, or sh Nsl1 interfering RNAs. Total histones were isolated by acidic extraction and H4K16ac, H4K5ac, and H4K8ac levels were analysed by western blot. Histones were normalized using an antibody against non-modified histone 3 (H3). KD efficiencies were tested in *Figure 2—figure supplement 2A–D*.

dramatic reduction of H4K16ac upon Msl1 depletion, whereas Nsl1 KD did not affect H4K16ac levels (*Figure 4D*). This is in good agreement with the differential Msl1 and Nsl1 ChIP-seq profiles (*Figure 4C*). Moreover, H4K5ac and H4K8ac levels did not change in cells expressing either Msl1 or Nsl1 shRNA (*Figure 4D*). Altogether, the above results indicate that in mESCs (i) the enzymatic activity of the MSL complex is responsible for H4K16 acetylation downstream of the TSS, (ii) MSL is the main acetylase for

H4K16 and (iii) the global H4K16 acetylating function of MSL cannot be compensated by other HAT complexes.

## MSL is present at genes regulating the pluripotency network and developmental processes

To understand the role of the two Mof-containing complexes for gene regulation and regulatory pathways in mESC, we further characterized genes bound either individually or together by Msl1 and/or Nsl1. Out of 10,600 Msl1- and Nsl1-bound genes about one quarter are co-bound by both complexes, while 3274 are only bound by Msl1 and 2185 only by Nsl1 (*Figure 5A*, *Figure 2—source data 2*). Our statistical analyses showed that these numbers are significant (*Figure 5—figure supplement 1*). To identify genes regulated specifically by MSL and/or by NSL, we analysed genes bound by only Msl1, by only Nsl1, or together by Msl1 and Nsl1 for gene ontology (GO). All three categories are enriched for GO terms such as metabolic process, gene expression, cell proliferation, and cell cycle. These GO terms represent housekeeping functions of every cell type, but can also be related to the cellular homeostasis of ESCs. Interestingly, genes bound by only Msl1 are enriched for GO terms such as embryo development, stem cell differentiation and maintenance (*Figure 5B*). Importantly, almost 50% of all reference genes associated with stem cell maintenance are Msl1 positive (*Figure 5B*).

Therefore, we investigated the presence of Msl1 (and Nsl1) at mESC-specific genes. Available RNA-seq data of mESCs (*Tippmann et al., 2012*) allowed us to define 282 genes expressed only in pluripotent mESCs. Out of these 282 mESC-specific genes, 123 (44%) are bound by Msl1 and only 40 (14%) are Nsl1 positive. Furthermore, about 100 mESC-specific genes are bound exclusively by Msl1, while 16 genes are bound only by Nsl1 (*Figure 5C*). Our statistical analyses indicate that only MSL binding at mESC-specific genes is higher than random (*Figure 5—figure supplement 2*).

To validate these bioinformatics analyses, all Msl1- and/or Nsl1-bound genes were divided into three categories (*Figures 2B and 5A*): Msl1- and Nsl1-bound genes (category 1), genes bound only by Nsl1 (category 2) and genes bound only by Msl1 (category 3). Msl1 and Nsl1 binding to the three gene categories were validated by ChIP-qPCR on a few selected genes. In agreement with all our above analyses, we observed that Msl1 binds to TSSs and/or GB regions of most genes from category 1 and 3 (*Figure 5D*), whereas Nsl1 is detected mostly at the TSSs of genes from category 1 and 2 (*Figure 5E*). Our results also show that Msl1 positive genes contain several mESC specific genes, including genes related to the core pluripotency network (e.g., Oct4, Nanog and Sox2) (*Figure 5E*). In summary, we demonstrate that the two Mof-containing complexes bind to shared, MSL-, or NSL-specific gene sets, but only the MSL complex is present at genes regulating the ESC pluripotency network and developmental processes.

## NSL influences cellular proliferation of mESCs

In mouse embryos ablation of *Mof* results in lethality at embryonic day 3.5 and Mof also affects mESCs pluripotency (*Gupta et al., 2008*; *Thomas et al., 2008*; *Li et al., 2012*). To further analyse the cellular roles of MSL and NSL in mESCs, Msl1, or Nsl1 were individually depleted by shRNA KD (see *Figure 2—figure supplement 2A–D*). To exclude compensation between MSL and NSL complexes, a double KD of Msl1 and Nsl1 (shMsl1/shNsl1) was also carried out (*Figure 6—figure supplement 1A*). Next, total cell numbers were counted over 6 days. These analyses indicated that KD of Msl1 reduces slightly cell proliferation, while the KD of Nsl1, or the double KD of Msl1 and Nsl1 lead to a much slower cell growth (*Figure 6A*). When analysing cell morphology under these KD conditions, we did not observe any change in mESC shape. (*Figure 6B*) As the reduction of cell numbers under the KD conditions was not due to apoptosis (*Figure 6—figure supplement 1B*), we next carried out cell cycle analyses. These FACS measurements demonstrated that mESCs treated with shMSL1, shNsl1 and shMsl1/shNsl1 accumulate in the G1-phase of the cell cycle, with shNsl1 and shMsl1/shNsl1 being more severe than shMsl1 (*Figure 6C*). These results together suggest that Nsl1 might be more required for regulating housekeeping genes involved in cellular homeostasis of mESCs, as reflected in higher number of G1-phase cells and decreased cell proliferation of shNsl1 and shMsl1/Nsl1 mESCs.

## MSL binds to pluripotency genes regulated by Mof

To better understand the function of genes bound by MSL and NSL in mESCs, genome-wide expression changes were analysed by microarrays. To this end total RNA was isolated from control mESCs, or mESCs depleted for either Msl1, or Nsl1 (*Figure 2—figure supplement 2A–D*). In shMsl1 KD cells, 275 genes were found to be downregulated (with Msl1 itself is in the downregulated list

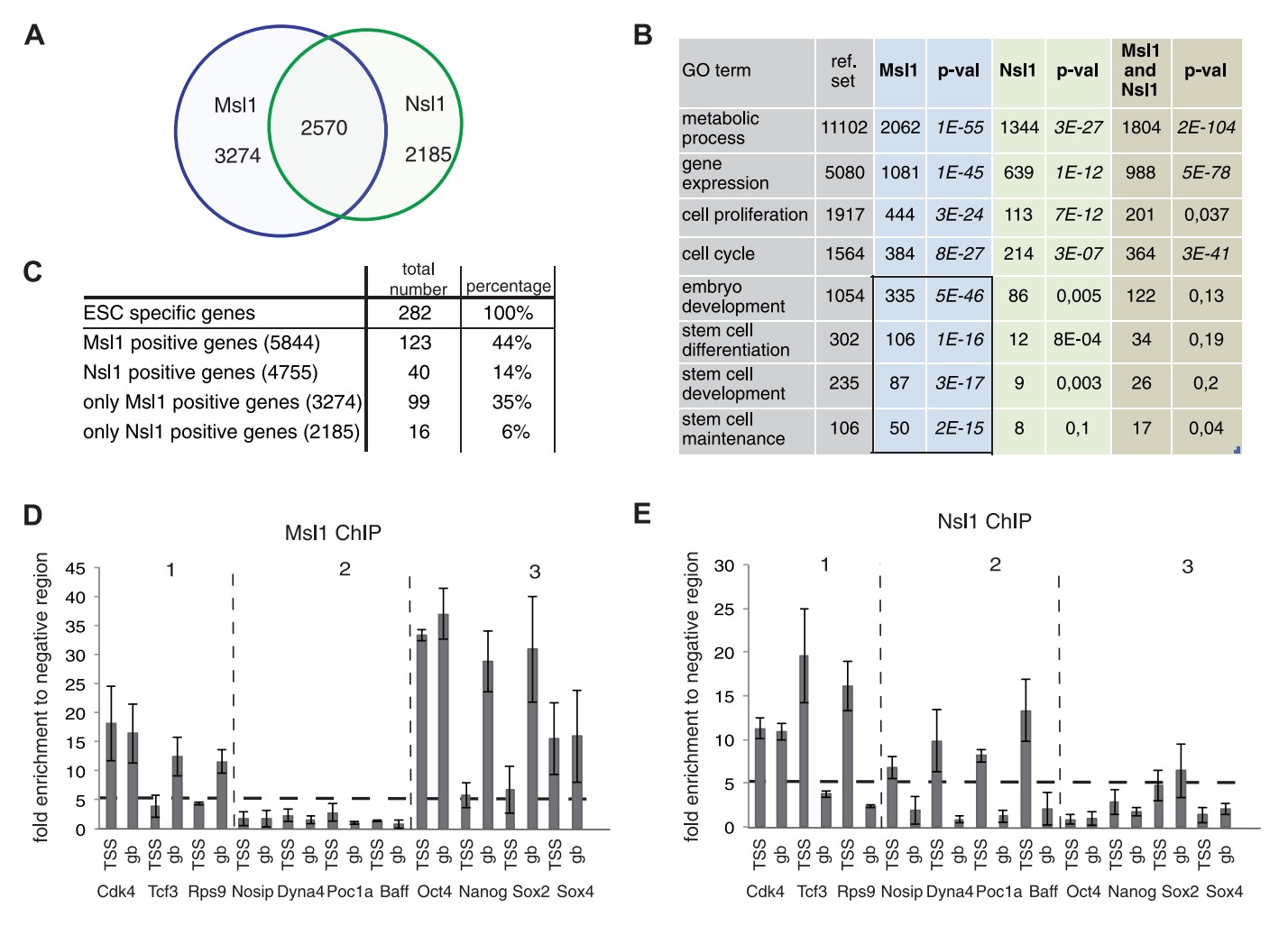

**Figure 5**. MSL and NSL bind to shared and specific gene sets. (**A**) Venn Diagram showing the overlap between Msl1 and Nsl1 binding sites at ENSEMBL genes. For statistical analyses see **Figure 5—figure supplement 1A–C**. Binding at TSSs and gene bodies was considered together. Genes are listed in **Figure 2—source data 2**. (**B**) Gene ontology analysis using Manteia (**Tassy and Pourquie, 2013**) of only Msl1 binding sites, only Nsl1 binding sites, or common binding sites. Significant GO terms for only Msl1 binding sites are highlighted by a box. (**C**) Differentially expressed genes were identified with the DEseq analysis (**Anders and Huber, 2010**). The table represents genes expressed only in mESCs (when compared to NPCs). The overlap with all Msl1 positive, all Nsl1 positive or only complex specific genes was calculated. Statistical analyses in **Figure 5—figure supplement 2A–D** indicates significant enrichment of Msl1 at mESC-specific genes. (**D** and **E**) Bound genes were divided into three categories: Common (category 1), specific for Nsl1 (category 2) and specific for Msl1 (category 3). From these categories genes were chosen for ChIP-qPCR using the anti-Msl1 (**D**) or anti-Nsl1 (**E**) antibodies. Fold enrichment higher than five was defined as specific binding and Msl1 and Nsl1 presence was analysed at the indicated genes in each category. Bar charts represent the mean and standard deviation of 2–3 independent experiments.

The following figure supplements are available for figure 5:

**Figure supplement 1**. MSL and NSL significantly bind to shared and specific gene sets.

**Figure supplement 2**. MSL, but not NSL locates to mESC-specific genes.

(**Figure 7—source data 1**)), and 500 genes upregulated, as compared to control KDs. By comparing Msl1-bound and Msl1-regulated genes (**Figure 7A**), we found that Msl1 is present at about 30% (105 genes) of all downregulated and at 20% (107 genes) of all upregulated genes. In Nsl1 KD conditions, 1158 genes are downregulated (including Nsl1 itself) and 429 are upregulated, as compared to KD controls (**Figure 7—source data 1**). By comparing the genome-wide binding (ChIP-seq) and expression data changes following KD, we show that Nsl1 is present at 43% (441 genes) of all

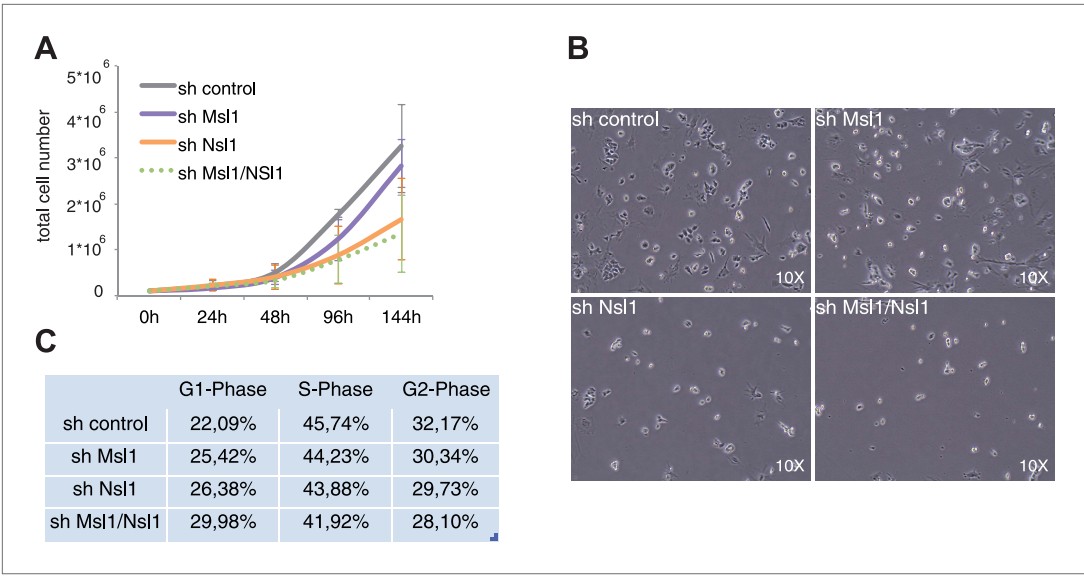

**Figure 6**. NSL influences cell growth and cell cycle of mESCs. (**A**) Cell proliferation analyses by cell counting over 6 days of control, or indicated KD mESCs. Error bars represent the standard deviation of three independent experiments. See *Figure 6—figure supplement 1A* for validation of KD efficiency of sh Msl1/Nsl1 double KD mESCs. (**B**) Morphology of control and Msl1 and Nsl1 single or double KD mESCs at 6 days after lentiviral infection using a reverse-phase microscope with a 10x magnification. (**C**) Cell cycle analyses of control and KD mESCs by propidium iodide staining followed by FACS analyses. Cell numbers of G1-, S- or G2-phases are represented in percentages after analyses with CellQuest Pro software. See *Figure 6—figure supplement 1B* for apoptosis analysis.

The following figure supplements are available for figure 6:

**Figure supplement 1**. shMsl1, shNsl1 and double KD mESCs do not undergo apoptosis.

downregulated and at 5% (30 genes) of all upregulated genes (*Figure 7B*). The Msl1-, or Nsl1-KD affected genes determined by the microarray analyses were then confirmed by RT-qPCR under Msl1-, or Nsl1-KD conditions (*Figure 7C,D*). Interestingly, we noticed genes known to be involved in differentiation, like *Nestin* and *Ntrk1*, in the upregulated genes of shMsl1 mESCs (*Figure 7C*).

Altogether, our results show that the bound genes of which the expression is affected following either Msl1 or Nsl1 KD are those genes, which absolutely require either MSL or NSL for their correct regulation. Note however, that the relatively weak overlap between Msl1- and/or Nsl1-bound genes on one side and Msl1- and/or Nsl1-regulated genes on the other, may reflect that either the KDs were not sufficiently efficient, and/or that the global gene expression analysis detected only changes in the steady state levels of the mature mRNAs and not the changes in the neosynthesized pre-mRNAs. Along these lines, in our experimental system Msl1, or Nsl1 single, or double KD mESCs do not loose Oct4 expression (*Figure 7—figure supplement 1A*), a common marker of mESC pluripotency.

Since MSL and NSL are transcriptional co-activators, we were interested in the biological function of downregulated genes upon KD of either Msl1 or Nsl1. We also included available expression data from *Mof* knock-out (KO) mESCs (*Li et al., 2012*) to overcome the above-described limitations. Analysing the biological function of Msl1, Nsl1, or Mof downregulated genes, we observed GO terms like metabolic processes, gene expression, cell death, or cell cycle control (*Figure 7—figure supplement 1B*). However, only downregulated genes in *Mof* KO mESCs are significantly enriched for GO terms like stem cell differentiation or maintenance (*Figure 7E*). Several of these genes are amongst the Msl1 positive mESC-specific genes, such as Nanog, Sox2, Oct4 (*Li et al., 2012*) (*Figure 7F*). Moreover, these genes are bound by Msl1 and Mof (*Figure 7—figure supplement 2*). Thus, the Mof-regulation and the exclusive binding of Msl1 and Mof to these key pluripotency genes suggest that the MSL complex is a regulator of the pluripotency network in mESCs.

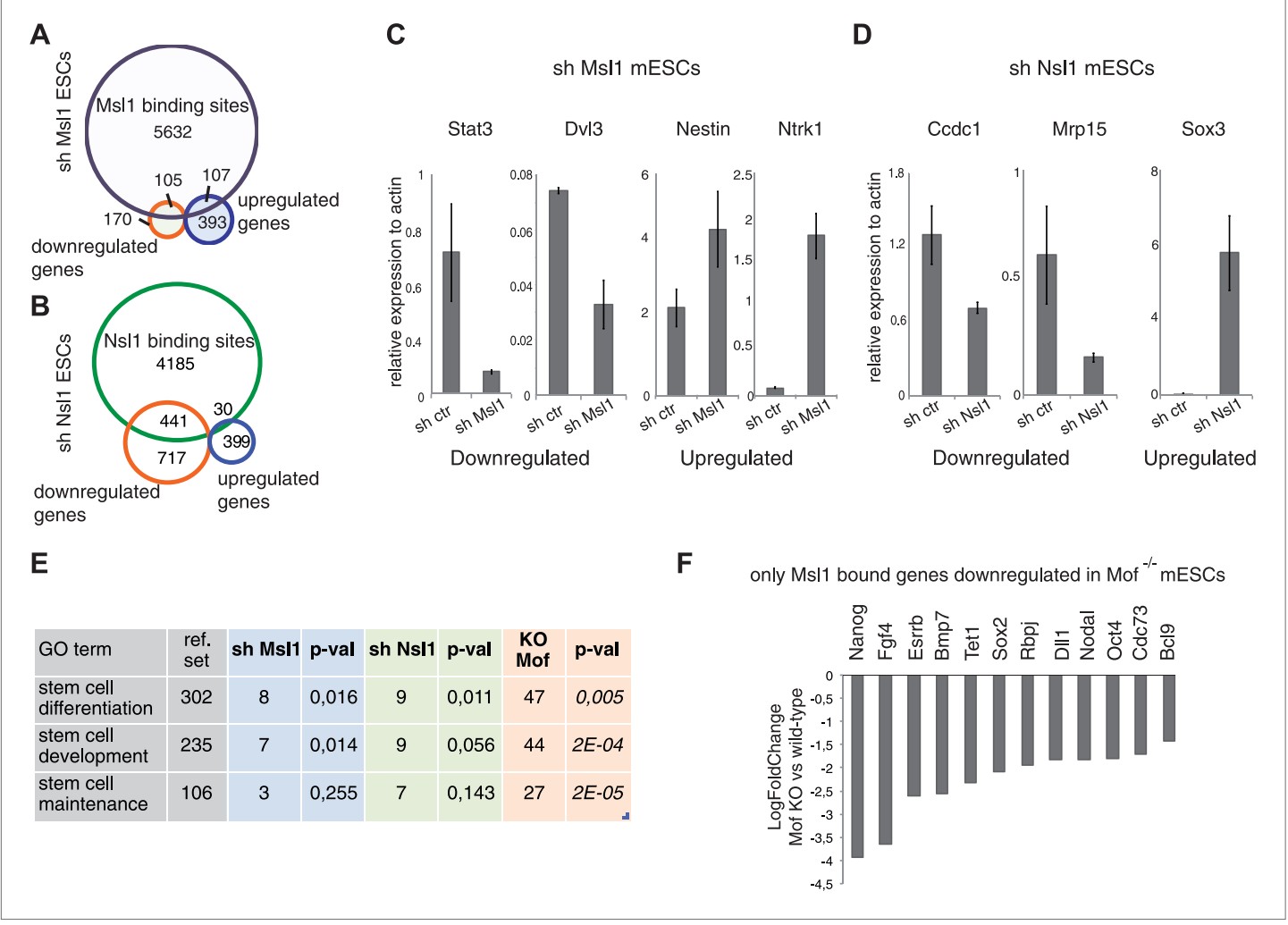

Figure 7. Analysis of Msl1 and Nsl1 regulated genes in mESCs. (**A** and **B**) Venn Diagram of Msl1 (**A**) or Nsl1 (**B**) MACS14 binding sites at ENSEMBL genes and down- or upregulated genes in Msl1 or Nsl1 KD cells. Down- and upregulated genes are listed in *Figure 7—source data 1*. For validation of KD efficiencies see *Figure 2—figure supplement 2A–D*. (**C** and **D**) RT-qPCR validation of down- and up-regulated genes in sh Msl1 (**C**) or sh Nsl1 (**D**) mESCs. (**E**) GO analyses using Manteia (*Tassy and Pourquie, 2013*) of all downregulated genes in Msl1 KD, Nsl1 KD and *Mof* KO mESCs. See *Figure 7—figure supplement 1* for further GO analyses. (**F**) Gene expression changes in *Mof* KO vs wild-type mESCs according to *Li et al. (2012)* of genes involved in stem cell maintenance, represented as fold change.

The following source data and figure supplements are available for figure 7:

**Source data 1**. Down- and upregulated genes in shMsl1 and shNsl1 mESCs.

**Figure supplement 1**. Msl1, Nsl1 and Mof regulate mESC-unspecific genes.

**Figure supplement 2**. Msl1 and Mof binding at pluripotency genes.

## MSL binds to bivalent genes in mESCs

Our above analyses have shown that Msl1 binds not only to mESC-specific genes, but that it locates also to silent, or very weakly expressed, genes that become expressed to control mESC differentiation (*Figure 8A*, *Figure 8—figure supplement 1A*). Importantly, KD of Msl1 leads to the upregulation of developmental genes, such as *Nestin* and *Ntrk1* (*Figure 7C*). These genes often contain both positive (H3K4me3) and negative (H3K27me3) epigenetic modifications (*Figure 8A*, *Figure 8—figure supplement 1A*). It is well established that H3K4me3 and H3K27me3 histone modifications co-localize at bivalent domains, which are poised for a quick activation during distinct differentiation processes

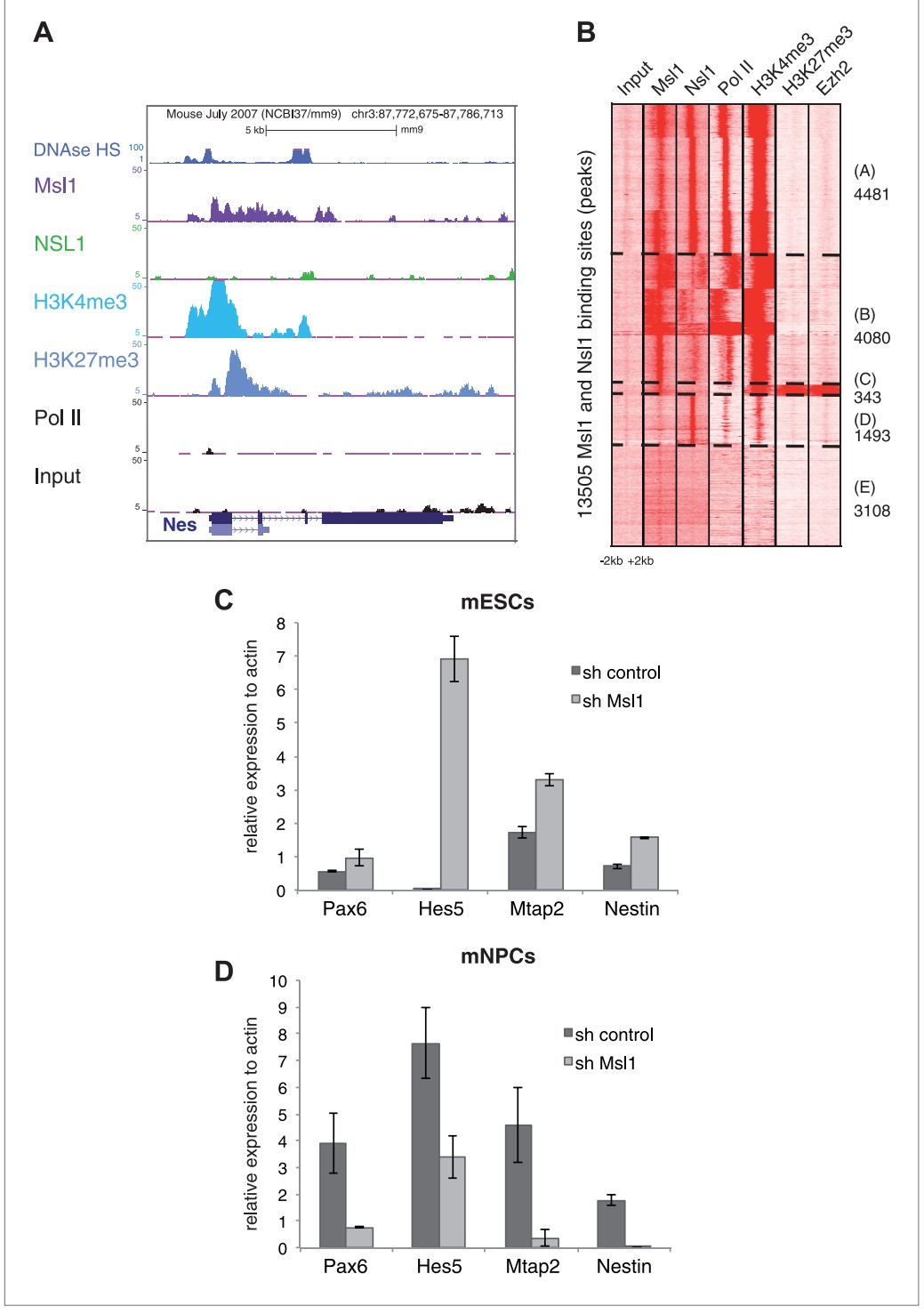

**Figure 8**. MSL regulates mESC differentiation. (**A**) Msl1 binding together with Nsl1, H3K4me3, H3K27me3 and Pol II at the *Nestin* locus in mESCs. For the *Hes5* gene locus see *Figure 8—figure supplement 1A*. (**B**) Heatmap showing k-means clustering of Msl1, Nsl1, Pol II, H3K4me3, Ezh2 and H3K27me3 using all Msl1 and Nsl1 binding sites as reference coordinates. Densities are presented −/+2 kb around reference coordinates. Based on the density profiles of all data sets, the heatmap is divided into different categories (as indicated). For statistical analyses of Msl1 positive bivalent genes in Cluster C see *Figure 8—figure supplement 1B*. (**C** and **D**) mRNA expression measurements by RT-qPCR of bivalent genes, which are also key markers for NPC differentiation, under sh control
*Figure 8. Continued on next page*

*Figure 8. Continued*

(dark grey) and sh Msl1 conditions (light grey) in pluripotent mESCs (**C**) or in mNPCs (**D**) NPC formation and KD efficiency of Msl1 in NPCs was validated in *Figure 8—figure supplement 1C,D*.

The following figure supplements are available for figure 8:

**Figure supplement 1**. MSL binds to bivalent genes.

(*Azuara et al., 2006*; *Bernstein et al., 2006*). The Ezh2 subunit of the polycomb repressive complex 2 (PRC2), which catalyzes histone H3K27 tri-methylation, is also a good marker of bivalent domains (*Bernstein et al., 2006*; *Ku et al., 2008*). To determine whether Msl1, or Nsl1, would bind to bivalent domains genome-wide, we compared the combined list of all Msl1 and Nsl1 binding sites with Pol II and Ezh2 profiles, together with H3K4me3 and H3K27me3 marks (*Figure 8B*). The heatmap indicates that about 343 Msl1 binding sites significantly co-localize with Ezh2, H3K27me3, and H3K4me3, which define the bivalent domains (see Cluster C in *Figure 8B*, *Figure 8—figure supplement 1B* for statistical analyses). Importantly, these 343 bivalent domain sites are negative for Nsl1 binding. The presence of Msl1 (our study) and Mof (*Li et al., 2012*) at bivalent genes in mESCs suggests that the MSL complex is involved in keeping these developmental genes silenced, but poised for activation in mESCs.

## MSL is required for gene regulation during mESC differentiation

As MSL, but not NSL, was found to be required for keeping bivalent genes silent or low expressed in pluripotent mESCs, we asked whether MSL could regulate bivalent gene expression during mESCs differentiation. For this, mESCs were differentiated into neuronal progenitor cells (NPCs) under control and Msl1 KD conditions (*Figure 8—figure supplement 1C*). Bivalent genes such as Pax6, Hes5, Mapt2 and Nestin, which are also considered as key markers of NPC differentiation, were upregulated in pluripotent mESC under Msl1 KD conditions (*Figure 8C*; and see above). In contrary, but in agreement of the regulatory role of the MSL complex at these genes, these key developmental marker genes were downregulated in NPCs in which Msl1 was silenced by shRNA expression during NPC differentiation (*Figure 8D*). Note however, that Msl1 KD cells morphologically are still able to form NPC-like cells (*Figure 8—figure supplement 1D*). These results together indicate the important regulatory requirement of the MSL complex first for keeping the subset of bivalent genes poised for activation in mESCs and then for turning them on during mESC differentiation.

## Discussion

### The composition and the genome-wide binding pattern of the MSL and NSL HAT complexes is conserved between mouse ESCs and other metazoan cells

In this study, we analysed two Mof-containing complexes, MSL and NSL to understand their transcription regulation function in mESCs. The proteomic characterization of the mMsl1-, or mNsl1-containing complexes indicated that the subunit composition of the mMSL and the mNSL complexes is identical to human complexes (*Figure 1C* and see *Mendjan et al., 2006*; *Cai et al., 2010*). Importantly, the comparison of the abundance of Msl1 or Nsl1 with Mof in the respective complexes (*Figure 1C*) and our gel filtration analyses (*Figure 1D*) demonstrated the incorporation of Msl1 and Nsl1 together with Mof in their respective endogenous complexes. Moreover, the gel filtration experiment indicated the potential existence of 'free' Mof that may not be present in either MSL or NSL. Together, the proteomic analyses suggest that there is no free Msl1 or Nsl1 in the nuclei of mESCs and that Msl1 and Nsl1 are specific to the MSL and NSL complexes, respectively.

This observation is important for our study as it indicates that the ChIP binding profiles obtained with either anti-Msl1 or anti-Nsl1 antibodies represent the behaviour of the corresponding endogenous Mof-containing MSL, or NSL HAT complexes. Furthermore, our findings are consistent with previous observations, which suggested that Msl1 and Nsl1 directly interact with Mof in their respective complexes, to stabilize the assembly of these complexes and to regulate their HAT activity (*Raja et al., 2010*; *Kadlec et al., 2011*).

Similarly to *Drosophila* (*Prestel et al., 2010*; *Raja et al., 2010*; *Kadlec et al., 2011*; *Feller et al., 2012*; *Lam et al., 2012*), our results demonstrate that mouse MSL and NSL have mainly distinct binding profiles at transcribed genes in mESCs. NSL binding overlaps with Pol II binding and DHSs at TSSs, while MSL locates more downstream of promoters towards the GB (*Figure 2*). These evolutionary conserved binding profiles further suggest that the function of the MSL and NSL complexes in transcriptional regulation are also conserved between *Drosophila* and vertebrate cells.

## MSL is the main H4K16 HAT in mESC

Surprisingly, in mESCs the KD of Msl1, but not that of Nsl1, leads to a global loss of H4K16ac, without reducing H4K5ac and H4K8ac levels (*Figure 4D*). In differentiated human cells Mof, and subunits of MSL (i.e. Msl1, Msl3) or the Nsl1 subunit of NSL have been reported to be crucial for global H4K16 acetylation by either MSL or NSL, respectively (*Li et al., 2009*; *Zhao et al., 2013*). Thus, while the subunit composition of the two Mof-containing MSL and NSL complexes is conserved between mESCs and differentiated human cells (*Figure 1C*; *Cai et al., 2010*), the function of the NSL complex seems to be differently regulated in pluripotent mESCs than in differentiated cells. Our observation that the KD of the Nsl1 subunit of NSL does not abolish global H4K16ac levels in mESCs suggests that in these pluripotent cells NSL may have a very localized HAT activity around TSSs of bound genes and the acetylation at these loci cannot be detected in total histone preparations, in contrary to the Msl1 KD. This difference may also be due to the more dynamic recruitment of NSL by mESC-specific factors. In contrast to MSL, NSL binding to promoters does not correlate with RNA expression levels (*Figure 3B–D*). This further suggests that the two Mof-containing complexes have different mechanisms of action in transcriptional regulation in mESCs. Moreover, the depletion of MSL function is supposed to recapitulate those chromatin perturbations and related cellular changes that are caused by the Mof KO and are linked to H4K16ac loss.

Strikingly, the gene expression of only a small number of MSL or NSL bound genes was directly affected when either Msl1 or Nsl1 was depleted (*Figure 7A,B*). This might be due to either inefficient KDs, or the measurement of steady-state mature mRNA levels under our experimental conditions. As contrary to Msl1 and Nsl1, 'free' Mof was detected in mESC nuclear extracts (*Figure 1D*), we cannot exclude the possibility that to certain extent Mof alone could compensate for the function of the MSL or NSL complex under Nsl1 and Msl1 KD conditions. However, the abolished global H4K16ac levels in shMsl1 mESCs would rather propose that other transcriptional co-activators, modifying other histone residues than H4K16, could compensate the role of H4K16ac in transcriptional activation.

In summary, our data demonstrating that MSL is the main HAT complex responsible for global H4K16ac in mESCs (*Figure 4D*), together with the finding that genome-wide binding profiles of Msl1 and Mof overlap with H4K16ac, (*Figure 2D*) suggest that the co-activator role of MSL is linked to its H4K16 acetylation function at the bound genes.

## MSL and NSL have different functions in pluripotent mESCs

The transitions between distinct chromatin states, from the open acetylated chromatin of the pluripotent mESCs to the more compact deacetylated chromatin of the differentiated cells, suggest the requirement for a tightly regulated chromatin acetylating/deactylating balance that participates in defining pluripotency on one hand and the consequent commitments for distinct differentiation pathways on the other hand. The HAT Mof is important for mESC pluripotency. Mof-deficient embryos have slight cell cycle defects and undergo cell death (*Thomas et al., 2008*). Under our experimental conditions, KD of Msl1 and Nsl1 alone or together did not affect expression of key transcription factors of the pluripotency network (*Figures 6A*, *Figure 7*). Our observation that in mESCs Nsl1 KD leads to decreased cell numbers during proliferation and an increase in cells in the G1-phase of the cell cycle shows that NSL might influence mESC proliferation (*Figure 6B–D*). This further indicates that NSL might be required for the homeostasis of mESC, either by directly regulating transcription or through acetylation of non-histone targets.

H4K16ac was shown to promote chromatin fibre decompaction in vitro (*Shogren-Knaak et al., 2006*; *Robinson et al., 2008*; *Allahverdi et al., 2011*). Our data showing that Msl1 KD abolishes global H4K16ac levels, together with the aberrant chromatin compaction observed in the Mof-deficient embryos (*Thomas et al., 2008*), suggest that the MSL complex is an important factor in establishing high acetylation levels required for more open chromatin conformation and consequent mESC pluripotency. In addition to its role as a general regulator of H4K16ac in mESCs, the MSL complex seems

to be recruited to ESC-specific loci to regulate different steps in the transcription process, such as (i) chromatin accessibility, (ii) pre-initiation complex formation and/or (iii) Pol II transcription elongation rates. Importantly, the Msl1-bound mESC-specific genes are regulated by Mof. Note however, these Mof- and Msl1-bound and Mof-regulated genes were not affected by the KD of Msl1 (*Figure 5B–D*, *Figure 7E,F*). As above explained this may be due to the different experimental systems used here and the *Mof* KO study. Nevertheless, we assume that these genes are regulated by the whole MSL complex. Altogether, the exclusive co-binding of Msl1 and Mof to pluripotency genes suggests that the MSL complex is a regulator of the pluripotency network in mESCs.

### MSL is required at bivalent genes to silence them in pluripotent ESCs and to upregulate them during ESCs differentiation

Bivalent genes, which are either repressed or expressed at very low levels in mESCs can be directly upregulated or completely silenced upon differentiation (*Azuara et al., 2006*; *Bernstein et al., 2006*). So far, little is known about the function of HATs at bivalent genes. Interestingly, we show that a subset of bivalent genes (about 350 genes) is bound by MSL in ESCs (*Figure 8A,B*, *Figure 8—figure supplement 1A*) and consequently that the KD of Msl1 results in the upregulation of a subset of bivalent genes in mESCs (*Figure 8C*). Consistent with our study, Mof has also been shown to be present at bivalent genes (*Li et al., 2012*). It seems that the MSL complex, probably in concert with HDACs and/or other chromatin remodelling factors, can have a silencing function at these bivalent genes. In contrast, the same genes require MSL for expression during differentiation (*Figure 8D*). Even though the morphology of NPCs was not obviously influenced under Msl1 KD conditions, expression of key developmental NPC genes, such as Pax6 and Hes5, were downregulated during NPC differentiation (*Figure 8D*). Thus, our findings together with the observation that Mof is also binding to bivalent genes in mESCs, strongly suggests that the presence of MSL at bivalent loci is important for keeping these bivalent genes poised in pluripotent mESCs, allowing a quick transcriptional upregulation of the same genes during mESC differentiation.

### Summary

In summary, MSL and NSL are key transcriptional co-activators at a large number of expressed genes in mESCs, whereas each complex has a distinct binding profile either at promoters (NSL) or gene bodies (MSL). MSL and NSL have overlapping and distinct roles in transcriptional regulation in mESCs. NSL binds mostly to genes with housekeeping functions and mediates mESC proliferation suggesting that NSL is important for the cellular homeostasis of mESCs. MSL is the main acetyltransferase complex acetylating H4K16. Moreover, MSL binds to mESC-specific genes, which are de-regulated in *Mof* ablated mESCs. Moreover MSL is present at bivalent domains in mESCs, where it may poise genes for activation during mESC differentiation. Importantly, expression of those genes is directly regulated by MSL in differentiated NPCs. In the future, it will be interesting to investigate how the genome-wide function of MSL and NSL changes during distinct mESC differentiation pathways.

## Materials and methods

### Cell culture

Wild-type male mESCs (E14.wt) were cultivated on 0.1% gelatine (Sigma, France) and CD1 feeder cells (37°C, 5% $CO_2$) in DMEM (4.5 g/l glucose) w-Glutamax-I, 15% foetal calf serum ESC-tested, leukemia inhibiting factor (5 µg) (Sigma), 50 mM ß-Mercaptoethanol (Invitrogen, France), penicillin/streptomycin (Invitrogen), 200 mM L-glutamine (Invitrogen), and non-essential amino acids (GIBCO, France). To work under feeder-free conditions cells were treated with 1 mg/ml Collagenase (GIBCO) and 2 mg/ml Dispase (GIBCO) and cultivated for one passage without feeder cells on 0.1% gelatine (Sigma) coated plates. Experiments were conducted at passage 26–29. Mouse embryonic fibroblasts (3T3 ATCC) were cultivated in DMEM (4.5 g/l glucose), 10% newborn calf serum and gentamycin (Invitrogen).

For NPC generation, we followed the protocol of *Bibel et al. (2007)*. Briefly, $6 \times 10^6$ mESC were cultured in DMEM (4.5 g/l glucose) w-Glutamax-I, 10% foetal calf serum ESC-tested, 50 mM ß-Mercaptoethanol (Invitrogen), penicillin/streptomycin (Invitrogen), 200 mM L-glutamine (Invitrogen), and non-essential amino acids on bacteriological Petri dishes (37°C, 5% $CO_2$) to start differentiation. After 4 days retinoic acid (5 µm) (Sigma) was added to induce NPC formation. Experiments were conducted 8 days after differentiation.

## Generation of antibodies

Polyclonal anti-Msl1 (3208) and anti-Nsl1 (3130) antibodies were generated by immunization of rabbits with the N-terminal (3-210 amino acids) region of mouse Msl1 or C-terminal region (762-1037 amino acids) of mouse Nsl1. The fragment was amplified and cloned in pET28b (Novagen, France) vector to express proteins in *E. coli* (BL21). For primer sequences see *Supplementary file 1*. Polyclonal antibodies were purified through Affi-Gel columns (Bio-Rad). For WB analysis anti-Msl1 (3208) or anti-Nsl1 (3130) antibodies were diluted 1:2000.

## Preparation of ESC nuclear extracts and immunoprecipitation

Nuclear extracts were prepared from 30 P15 plates of mESCs with 80% confluency as described in *Demeny et al. (2007)*. Proteins of 3 mg (Msl1) or 1 mg (Nsl1) mESC nuclear extracts were immunoprecipitated (IP) with 100 µl protein A Sepharose beads and 20 µl of the anti-Msl1 (3208) or 20 µl of the anti-Nsl1 (3130) antibody. Antibody-protein A Sepharose containing the bound proteins were washed three times with IP buffer (25 mM Tris-HCl pH 7.9, 10% glycerol, 0.1% NP40, 0.5 mM DTT, 5 mM $MgCl_2$) and 100 mM KCl and afterwards with IP buffer containing 250 mM KCl. Proteins were eluted from protein A Sepharose beads 150 µl of 0.1 M Glycine pH 2.6. Elutions were neutralized by adding 50 µl of 2 M Tris pH 8.5.

## MudPIT analysis

MudPIT analyses were performed as previously described (*Washburn et al., 2001*; *Florens et al., 2006*). In summary, protein mixtures were TCA precipitated, urea-denatured, reduced, alkylated, and digested with endoproteinase Lys-C (Roche) followed by modified trypsin digestion (Promega). Peptide mixtures were loaded onto a triphasic 100 µm diameter fused silica microcapillary column described as follows (*McDonald and Yates, 2002*). Loaded microcapillary columns were placed in-line with a Quaternary Dionex Ultimate 3000 HPLC pump and a LTQ Velos linear ion trap mass spectrometer equipped with a nano-LC electrospray ionization source (ThermoFischerScientific). A fully automated 12-steps MudPIT run was performed as previously described (*Florens et al., 2006*) during which each full MS scan (from 300 to 1700 m/z range) was followed by 20 MS/MS events using data-dependent acquisition. Proteins were identified by database searching using SEQUEST (*Eng et al., 1994*) within ThermoProteome Discoverer 1.3 and 1.4 (ThermoFischerScientific). Tandem mass spectra were searched against a *Mus musculus* protein sequence database containing 16,604 entries (from the Swissprot 2013-04-03 release). In all searches, cysteine residues were considered to be fully carboxyamidomethylated (+57 Da statically added) and methionine considered to be oxidized (+16 Da dynamically added). Proteins were considered as specific in a given IP data set if they were absent or 10-fold minimum enriched as compared to a MOCK IP, performed on the same protein input by using a non-specific antibody targeting yeast TAF90. Relative protein abundance for each protein in either the anti-Msl1, or the anti-Nsl1 IPs was estimated by the calculation of a Normalized Spectral Abundance Factor (NSAF) (*Zybailov et al., 2006*). NSAF values were calculated from the spectral counts of each identified protein. To account for the fact that following enzymatic digestion larger proteins result in more peptides/spectras than small proteins, each given spectral count was divided by the corresponding protein length to provide a spectral abundance factor (SAF). To obtain NSAF, SAF values were normalized against the sum of all SAF values in the corresponding run. Thus, NSAF values obtained from a given protein mixture, such as immunoprecipitated protein complexes, allow the comparison of the abundance of a given protein/subunit to another in the same mixture/complex.

## Gel filtration

For gel filtration a Superose 6 (10/300) column pre-equilibrated in 25 mM Tris pH 7.9, 1 mM DTT, 5 mM $MgCl_2$, 150 mM KCl, and 5% Glycerol was used. 250 µl calibration mix containing Dextran Blue (2 MDa) and Biorad calibration kit (ref 151-1901) with marker sizes of 670 kDa, 158 kDa, 44 kDa, 17 kDa, and 1.35 kDa were injected at 0.3 µl/min. 500 µl of mESC nuclear extract containing 1 mg protein was injected and run at 0.3 µl/min. 40 fractions were collected and analysed by western blot. For western blot the anti-Mof (A300-992a; Bethyl) antibody was used.

## Chromatin immunoprecipitation

ChIP was carried out as described previously with slight modifications (*Krebs et al., 2011*). At 80% confluency mESCs were cross-linked with 1% formaldehyde for 10 min at room temperature, lysed and shared mechanically using the Covaris E210 to obtain a chromatin fragment size of 200–500 bp. IP were carried out using 500 µg of chromatin. For the IP 3 µg of purified Msl1 3208 or Nsl1 3130 antibodies were

used. The input was obtained from 50 μg of chromatin, pre-cleared, and directly reverse crosslinked. DNA was purified using a Qiaquick (Qiagen, France) column. Quantitative real-time PCR (qPCR) was performed with SYBR Green (Roche). Primer sequences are summarized in the *Supplementary file 1*.

## ChIP-seq

10 ng of precipitated DNA obtained from ChIP was used for Solexa sequencing. To create a genomic library, we followed the instructions of NEXTFlex v12.03 (BIO Scientific) for Msl1 and the NEBNext protocol (E6240; Biolabs) for Nsl1. Libraries were validated with the Agilent Bioanalyzer. Single reads run sequencing was conducted with the HiSeq 2000. Image analysis and base calling were done with the Illumina pipeline (1.8.2). The July 2007 *Mus musculus* genome assembly (NCBI37/mm9) from NCBI was used for the sequence alignment by the software Bowtie (0.12.7) (*Langmead, 2010*). All analyses were conducted with unique reads. Bed files were used to create read density (wig) files by extending reads to 200 bp length and creating 25 bp bins. We further included following sequencing datasets, which were obtained from Gene Expression Omnibus (www.ncbi.nlm.nih.gov/geo/) in our analysis: Input (GSM798320) (*Karmodiya et al., 2012*), RNA Polymerase II (GSM307623), H3K4me3 (GSM307618), H3K27me3 (GSM307619), Ezh2 (GSM327668) (*Mikkelsen et al., 2007*), H4K16ac (GSM1156617) (*Taylor et al., 2013*), and Mof (GSM915227) (*Li et al., 2012*). Fastq files were generated from SRA lite format and aligned to the NCBI37/mm9 assembly using Bowtie (0.12.7) (*Langmead, 2010*). DHS were obtained from Encode/UW (GSM1014154).

## Peak detection and annotation

To detect Msl1 and Nsl1 peaks, the algorithm MACS14 (*Zhang et al., 2008*) was applied using default parameters with slight modifications. For Msl1 peak detection the p-value cutoff was set to $10^{-5}$, no shifting model was built and the shift size was defined as 200. The annotation was based on the ENSEMBL 67 database (mm9). Peaks were annotated to genomic features (TSS, TTS, CDS Exons, 5'UTR, 3'UTR, Introns and intergenic) using the software HOMER (4.2) (*Heinz et al., 2010*) with default parameters.

## Data analysis

To calculate the Msl1, Nsl1, or Pol II enrichment at a given gene either the peak tag density of the nearest peak to the TSS (in a region of +2 kb), was obtained through MACS14 (*Zhang et al., 2008*) or the total tag density around the TSS (+2 kb) was taken. Further analysis and graphical representation were conducted using the software R.

Density profiles around the TSS and GB were obtained through seqMINER (*Ye et al., 2011*). For the comparison and analysis of genomic features between data sets, the software BEDTools (2.17.0) (*Quinlan and Hall, 2010*) was used. Scatter plots and Pearson correlations with Pearson p-values were obtained by calculating the log2 values of read densities normalized to the control at the given peaks or around ENSEMBL transcription start sites. K-means linear clustering was conducted and represented with seqMINER. Venn diagrams were generated with Biovenn (*Hulsen et al., 2008*). Manteia (v.2) was used for GO analysis of batch gene entries to understand the biological function (*Tassy and Pourquie, 2013*). Only GO levels between 1 and 10 were taken into consideration and compared between groups.

## Bootstrap statistical analysis

To verify the statistical significance of the obtained Msl1- or Nsl1-bound gene groups in *Figure 5A,C* and *Figure 8B*, we performed bootstrap statistical analyzes for *Figure 5A,C* and *Figure 8B*. In all these analyses, we used the total pool of 26,460 ENSEMBL genes. Next out of these pools, we randomly selected the same number of total events (genes or binding sites) than those determined non-randomly in the corresponding figures (i.e., 10600 in *Figure 5A*; 282 in *Figure 5C* and 13,505 in *Figure 8*). This random selection was then compared with the different given interest gene lists (i.e. 3274, 2570, and 2185 for *Figure 5A*) and the number of genes (IDs) belonging to the non-random experimental group was determined. We repeated this process of random selection and gene list crossings 10,000 times and represented the number of IDs and their observed frequencies as histograms (see corresponding figure supplements). For each gene list, we computed an average (mean) and a standard deviation (sd) of the number of random matches. A z-score is computed as: z = (mean-expect)/sd, where 'expect' is the number of expected interest genes. p-values associated to these scores are indicated in the corresponding figure legends. On each histogram we indicated in bold the number of IDs found in the non-random experimental group. The p-value represents the significance of the difference between the randomly found average and the experimental ID numbers.

## RNA-seq

Gene expression levels are based on the ENSEMBL 67 database (mm9). Raw data of mESCs and NPCs were taken from Gene Expression Omnibus (GSE34473) and processed using the software tools TopHat (*Trapnell et al., 2009*) and HTSeq with default parameters. FPKM (fragments per kilobase of exon per million fragments mapped) values were calculated with Cufflinks (*Roberts et al., 2011*). Differentially expressed genes (DE) in mESCs and NPCs were identified with the bioconductor package DESeq (1.14.0) (*Anders and Huber, 2010*) using default parameters.

## shRNA interference

shRNA approaches were conducted with pLKO.1 puro shRNA vectors (Sigma–Aldrich, France) of the TRC2 library. For Ns1 KD the TRCN0000241466 shRNA clone and for Msl1 KD the TRCN0000241378 shRNA clone was used. Double KD of Msl1 and Nsl1 was conducted with equal amount of the TRCN0000241466 and TRCN0000241378 shRNA clones. For control the shRNA non-target control (Product No. SHC002) was applied. Production of lentiviral particles as well as infection of mESCs was conducted according to the manufacturer's protocol. 3 days after viral transfection of $2 \times 10^6$ mESCs selection with puromycin (2 μg /ml) (InvivoGen) was started. Experiments were conducted 5 days after viral transfection. KD efficiency was tested at RNA levels through reverse transcriptase (RT)-qPCR (see *Supplementary File 1*) and at protein levels through western blot of whole protein extracts. Moreover, mRNA expression of selected genes was analysed by (RT)-qPCR, whereas primer sequences are summarized in *Supplementary file 1*. Total RNA, which was used for gene expression profiles and cDNA synthesis, was isolated with TRIzol reagent (Invitrogen) and treated with DNAse. cDNA was synthesized with Transcriptor reverse transcriptase (Roche) using random hexamers according to the manufacturer's protocol. For normalization of protein amount by WB analyses the anti-Tubulin (T6557; Sigma-Aldrich) antibody and ponceau solution (Sigma-Aldrich) was applied. To analyse the pluripotency state of KD mESCs the anti-Oct4 (611202; BD Labs) antibody was used. To analyse cell morphology images were taken with the digital inverted EVOS XL core (Fisher Scientific, France) microscope using a 10X objective.

## Isolation of mESC total histones and western blot assay

Histones were prepared from mESCs by lysing cells in 10 mM HEPES, pH 7.5, 1.5 mM $MgCl_2$, 10 mM KCl, 0.5 mM DTT, 100 mM Natrium Butyrate, and 0.2 M HCl for 30 min on ice, centrifuged and dialysed first against 0.1 M acidic acid and then against water. Samples were analysed by western blot for histone modifications using the anti-H4K16ac (07-329; Millipore, France), anti-H4K5ac (51997; Abcam, UK), anti-H4K8ac (15823; Abcam), and anti-H3 (1791; Abcam) antibody.

## Cell growth analysis

Cell growth analyses was started 6 days after lentiviral infection by plating $1 \times 10^5$ mESCs on 0.1% gelatine coated per 6-well plates in triplicates. mESCs were counted in triplicates using Neubauer cell counting chambers at indicated time points. mESCs were split every second day to $1 \times 10^5$ mESCs/6-well.

## Cell cycle analysis

7 days after lentiviral infection $5 \times 10^5$ mESCs were dissolved in 1 ml PBS (0.1% NaCitrate and 0.1% TritonX 100). Propidium iodide (50 μg/ml) was added and after 4 hr incubation on ice cells were analysed by the FACS calibur. Data were analysed using the CellQuest software.

## Cell death assay

Cell death was examined using the APOPercentage apoptosis assay (A1000/DC79; Biocolor, France) following the manufacturer's instructions. As a positive control apoptosis was induced with 10 mM hydrogen peroxide for 8 hr in sh control cells. Absorbance was read at 550 nm and normalized to the blank control (without cells).

## Gene expression profiles and statistics

Experiments were designed with three independent biological replicas. Biotinylated cDNA targets were prepared, starting from 150 ng of total RNA, using the Ambion WT Expression Kit (Cat 4411974), and the Affymetrix GeneChip WT Terminal Labelling Kit (Cat 900671) according to Affymetrix recommendations. Following fragmentation and end-labeling, 3 μg of cDNAs were hybridized on GeneChip

Mouse Gene 2.0 ST arrays (Affymetrix, UK) for whole-transcript expression profiles. Washed and stained chips were scanned with the GeneChip Scanner 3000 7 G (Affymetrix) at a resolution of 0.7 µm. Obtained raw data (.CEL intensity files) were processed with Affymetrix Expression Console software version 1.1 to calculate probe set signal intensities using Robust Multi-array Average (RMA) algorithms with default settings.

To select the DE genes, we used the fold change rank ordering statistics (FCROS) method (*Dembele and Kastner, 2014*). In the FCROS method, k pairs of test/control samples are used to compute fold changes (FC). For each pair of test/control samples, obtained FCs for all genes are ranked in increasing order. Ranks that result are associated to genes. Then, the k-ranks of each gene are used to calculate a statistic, and resulting probability (f-value) is used to identify the DE genes with an error level of 5%.

## Accession numbers

Msl1 and Nsl1 ChIP-seq data sets as well as gene expression profiles of sh control, sh Msl1 and sh Nsl1 mESCs are deposited at Gene Expression Omnibus (www.ncbi.nlm.nih.gov/geo/) under the accession numbers: GSE53797 and GSE56646.

## Acknowledgements

We are grateful to JW Conaway, ME Torres Padilla, V Pavet-Portal and D Langer for materials, helpful discussions and advice. We thank F Klein for the help in the gelfiltration experiment, M Gerard, D Devys, and A Krebs for critically reading the manuscript and for helpful comments, the IGBMC microarray and sequencing platform data generation and bioinformatics support; G Duval for antibody generation; the mass-spectrometry facility; M Hestin and G Rossi for help in ESC culturing. SR was supported by a fellowship from ARC. This work was supported by funds from CNRS, INSERM, Strasbourg University, and ANR (ANR-09-BLAN-0266; ANR-09-BLAN-0052) grants. This study was also supported by the grant ANR-10-LABX-0030-INRT, under the frame programme Investissements ANR-10-IDEX-0002-02.

## Additional information

### Funding

| Funder | Grant reference number | Author |
|---|---|---|
| Agence Nationale de la Recherche (L' Agence Nationale de la Recherche) | ANR-09-BLAN-0266; ANR-09-BLAN-0052, Investissements ANR-10-IDEX-0002-02: ANR-10-LABX-0030-INRT | Làszlò Tora |

The funder had no role in study design, data collection and interpretation, or the decision to submit the work for publication.

### Author contributions

SR, Conception and design, Acquisition of data, Analysis and interpretation of data, Drafting or revising the article; MF, TY, DD, VC, Acquisition of data, Analysis and interpretation of data; MS, Conception and design, Acquisition of data; LT, Conception and design, Analysis and interpretation of data, Drafting or revising the article

### Author ORCIDs

Doulaye Dembele, http://orcid.org/0000-0003-3879-6940

## Additional files

**Supplementary file**
• Supplementary file 1. The table contains all primer sequences used for cloning of Msl1, or Nsl1 cDNA fragments into the pET28b expression vector, used for Msl1 and NSl1 ChIP-qPCR and RT-qPCR to analyse gene expression profiles.

## Major datasets

The following datasets were generated:

| Author(s) | Year | Dataset title | Dataset ID and/or URL | Database, license, and accessibility information |
|-----------|------|---------------|----------------------|--------------------------------------------------|
| Ravens S, Tora L | 2014 | MOF-associated complexes have overlapping and unique roles in regulating pluripotency in embryonic stem cells and during differentiation | http://www.ncbi.nlm.nih.gov/geo/query/acc.cgi?acc=GSE56646 | Publicly available at NCBI Gene Expression Omnibus. |
| Ravens S, Tora L | 2014 | MOF-associated complexes have overlapping and unique roles in regulating pluripotency in embryonic stem cells and during differentiation GSM1300939 Msl1_ChIP-seq; GSM1300940 Nsl1_ChIP-seq) | http://www.ncbi.nlm.nih.gov/geo/query/acc.cgi?acc=GSE53797 | Publicly Available at NCBI Gene Expression Omnibus. |

**Standard used to collect data:** GEO Submissions (GSE56646 and GSE53797) [NCBI tracking system #17012666] data was submitted and curated according to the standards of GEO database rules.

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
