## [Decision Letter]

Thank you for sending your work entitled “MSL and NSL HAT complexes have overlapping and distinct roles, with MSL being the embryonic stem cell-specific regulator” for consideration at *eLife*. Your article has been evaluated by a Senior editor, a Reviewing editor, and 2 reviewers.

The Reviewing editor and the reviewers discussed their comments before we reached this decision, and the Reviewing editor has assembled the following comments to help you prepare a revised submission.

The reviewers were generally supportive of the paper but felt that there were still some fairly significant issues that would need to be addressed before the paper could be considered for publication. Most important relate to the biological functions of the complexes. What is the phenotype of the Msl1 or Nsl1 knockdown and how does this compare to Mof knockdown/depletion? Is all the Mof really in the NSL or MSL complex, as you claim? If you can address the reviewers’ comments (summarized below) in a timely manner we would be happy to consider a revised version.

1) Genetic mutation of Mof has a lethal phenotype in mice and ES cells. The authors apply depletion of Msl1 and Nsl1 but do not seem to provide data on how these affect stem cell behaviour. Self-renewal and differentiation potential need to be analysed carefully. Also if lethality is caused by loss of either complexes (NSL or MSL, the latter more likely seems to be the major factor in ES cells) then changes in gene expression could be due to cell death. This needs to be carefully considered and discussed. Alternatively, does combined Msl1 and Nsl1 depletion recapitulate the Mof mutation?

2) In Figure 2 H4K16as shows peaks (right most H4K16ac peak) where neither Nsl1 nor Msl1 are enriched. This indicates that other complexes also act. A Mof chromatin IP would be useful to see if H4K16ac overlaps with Mof as would be expected.

3) The authors conclude that there is no free Nsl1 and Msl1 protein as all would associate in Mof complexes. I wonder if the data allows this conclusion. To make this point the authors would need to demonstrate that their measurements would in principle allow detecting 10% free Msl1 or Nsl1 protein. To conclude from “Msl1 or Nsl1, incorporate in their respective complexes with similar abundance as does Mof” is too vague to make this point.

4) The overlap between MSL/NSL and bivalent genes appears not very significant (Figure 6 – only cluster C contains bivalent genes). The large majority of Msl1 and Nsl1 bound genes are not H3K27me3. Throughout the manuscript genes are categorized but statistical testing seems to be not performed. Rigorous statistical tests and control groups need be included to establish meaningful correlations.

5) Acetylation of other residues than H4K16ac is suggested (Discussion) as a redundant mechanism. It is not clear how to reconcile this speculation fits with the Mof phenotype.

---

## [Author Response]

1) Genetic mutation of Mof has a lethal phenotype in mice and ES cells. The authors apply depletion of Msl1 and Nsl1 but do not seem to provide data on how these affect stem cell behavior. Self-renewal and differentiation potential need to be analysed carefully. Also if lethality is caused by loss of either complexes (NSL or MSL, the latter more likely seems to be the major factor in ES cells) then changes in gene expression could be due to cell death. This needs to be carefully considered and discussed. Alternatively, does combined Msl1 and Nsl1 depletion recapitulate the Mof mutation?

As required we have included following experiments in our revised manuscript to address the reviewers’ first points:

a) To analyse whether Msl1 and Nsl1 play a role in maintenance of ESC pluripotency, we have conducted single Msl1 (shMsl1), or Nsl1 (shNsl1) or combined of Msl1 and Nsl1 (shMsl1/Nsl1) knockdowns (KDs) in mESCs (see new Figure 6—figure supplement 1). First we determined total cell numbers of ESCs, in which we depleted either Msl1, or NSL1, or both. Under these conditions we observed that especially shNsl1 and shMsl1/Nsl1 ESCs had a much slower cell proliferation rate, when compared to control ESCs, whereas the morphology of these ECS was not altered (see new Figure 6). Next we measured whether the observed reduction of cell numbers was due to apoptosis. However, we did not find any increase in apoptotic cells under these KD conditions (new Figure 6—figure supplement 1). Therefore next we tested whether the KD ESCs would be blocked in any particular cell cycle phase by using FACS analyses (new Figure 6). In agreement with the above observed cell numbers, we found that KD ESCs accumulated in G1-phase of the cell cycle and that this increase was more severe in shNsl1 and shMsl1/Nsl1 ESCs. We have further described these new observations in the Results section “NSL influences cellular proliferation of mESC”.

b) Note, however, that we do not observe detectable changes in Oct4 (new Figure 7—figure supplement 1) and alkaline phosphatase (data not shown) protein levels in shMsl1, shNsl1 and shMsl1/shNsl1 KD mESCs. As described in the Results and Discussion sections of our revised version, we cannot exclude that the lentiviral KDs are only partially efficient or the compensation of MSL and/or NSL function by “free” Mof (see also new Figure 1 and our answer to point 3). Even though the pluripotency state does not seem to be affected (as judged by the expression of the used markers) and the double knockdown does not entirely recapitulate the Mof KO phenotype, our new results suggest that Nsl1 is more required for regulating housekeeping genes involved in cellular homeostasis of mESCs. As required these new observations are further discussed in the revised manuscript.

c) Since we observed MSL binding at developmental bivalent genes (Figure 5 and Figure 8), which are also upregulated in shMsl1 mESCs (new Figure 7), in agreement with the reviewers question, we have further analysed the differentiation potential of mESCs depleted for Msl1. For this, mESCs were differentiated into neuronal progenitor cells (NPCs) under control and Msl1 KD conditions (see new Figure 8—figure supplement 1). First, we analysed expression profiles of bivalent as well as developmental genes, such as Pax6, Hes5, Mapt2 and Nestin, which are also considered as key markers of NPC differentiation. Importantly, our new results show that while these key developmental marker genes, including several bivalent genes, become upregulated in pluripotent mESC under Msl1 KD conditions (new Figure 8), their expression are in contrary downregulated in NPCs in which Msl1 was silenced during NPC differentiation (new Figure 8). Note however, that Msl1 KD cells morphologically are still able to form NPC-like cells (Figure 8–figure supplement 8D). These new results indicate the important regulatory requirement of the MSL complex for the expression of bivalent genes, known to become upregulated during cellular differentiation, in mESC and further differentiated NPCs. These new observations and figures are described in the Results section and further discussed in the Discussion section.

In conclusion, we show that Nsl1 regulates proliferation and cellular homeostasis of mESCs. Knockdown of Msl1 leads to a global loss of histone H4K16ac indicating that MSL is the main HAT acetylating H4K16 in mESCs. MSL is enriched at many mESC-specific genes, but also at bivalent domains. Interestingly, MSL is important to keep a subset of bivalent genes silent in pluripotent ESCs, while the same genes require MSL for expression during differentiation. In agreement, during neuronal differentiation MSL is essential for the regulation of key developmental genes.

*2) In*
Figure 2
*H4K16as shows peaks (right most H4K16ac peak) where neither Nsl1 nor Msl1 are enriched. This indicates that other complexes also act. A Mof chromatin IP would be useful to see if H4K16ac overlaps with Mof as would be expected*.

By using the commercially available anti-MOF antibody batches we obtained only very low enrichment values by ChIP-qPCR at defined genomic loci (such as the region shown in Figure 2). The differences between our Mof ChIP-qPCR results and the available Mof ChIP-seq data might be due to different experimental setups or antibody batches.

Thus, as suggested we included the published Mof ChIP-seq results (Li et al. (2013) in our analyses (see new Figure 2). Figure 2 shows that the “right most H4K16ac peak” (with a very low tag density) in the Cdk19 gene overlaps with Mof binding. However, at this specific loci neither Nsl1, nor Msl1 can be detected. This can be explained by several ways: a) the “free” Mof detected in ESC nuclear extracts (see new Figure 1 and our answers to point 3) is binding at these sites, b) the anti-mMof antibody used in the ChIP-seq study, in addition to mouse Mof, is also recognizing another protein (with a similar epitope) that may also bind to DNA, or c) due to their specific conformation and/or involvement in special chromatin structures neither MsL1, nor Nsl1 can be crosslinked at these sites. Along the same lines, Straub et. al. (2013 Genome Research), when comparing ChIP-chip and ChIP-seq profiles, suggested that *Drosophila* Msl1 and Mof might not be detectable by ChIP-seq approaches in genebodies due to the extensive fragmentation of chromatin (to obtain very small DNA fragments (200 bp)), which are then used for sequencing. Thus, extensive fragmentation might result in disruption, or at least partial disruption, of large chromatin-bound complexes, resulting in ChIP-seq signal loss of proteins not directly associated with the chromatin. If Mof would be closer to the DNA than Msl1, or Nsl1, respectively, in MSL or NSL complexes at certain loci, the sole detection of Mof could be explained by the suggestion of Straub et al (2013).

Importantly however, our new Figure 2 together show that the genome-wide Mof binding profile at promoters and in genebodies is very similar to Nsl1 (at promoters) or Msl1 (at regions downstream from promoters) binding (Figure 2 and Figure 2).

These new results are now described in the Results section.

*3) The authors conclude that there is no free Nsl1 and Msl1 protein as all would associate in Mof complexes. I wonder if the data allows this conclusion. To make this point the authors would need to demonstrate that their measurements would in principle allow detecting 10% free Msl1 or Nsl1 protein. To conclude from “Msl1 or Nsl1, incorporate in their respective complexes with similar abundance as does Mof” is too vague to make this point*.

We apologize if our conclusion obtained from our mass spec results (Figure 1) were not well explained. Therefore, we improved the description of the meaning of the NSAF abundance calculations in our revised version (see Results section and Materials and methods).

Briefly, the development of non-gel-based, “shotgun” proteomic techniques such as Multidimensional Protein Identification (MudPIT) has provided powerful tools for studying large-scale protein characterization in complex biological systems. As during enzymatic digestions of protein mixtures for proteomic analyses large proteins contribute more peptide/spectra than small ones, a normalized spectral abundance factor (NSAF) was defined to account for the effect of protein length on spectral count for comparing protein abundance in the different samples ([60], Journal of Proteome Research; [15], Methods). NSAF is calculated as the number of spectral counts (SpC) identifying a protein, divided by the protein’s length (L), divided by the sum of SpC/L for all proteins in the experiment. Thus, NSAF allows the comparison of abundance of individual proteins in multiple independent samples and has been applied to quantify the subunit abundance in various protein mixtures and in multiprotein complexes ([15], Methods; Paoletti et al. 2006, PNAS; Bieniossek et al. 2013, Nature). This NSAF counting method provides an easy way of identifying proteins with similar (or different) abundances in immunoprecipitated (IP-ed) protein complexes using MudPIT. This is the method we used. As the NSAF values of Msl1 (7,8) and Mof (13,5), or Nsl1 (9,3) and Mof (14), were in a comparable range (see Figure 1) in either the anti-Msl1, or the anti-Nsl1 protein immunoprecipitations, we concluded that all the IP-ed Msl1, or Nsl1 associated with comparable amounts of Mof in mESCs, respectively. If free Msl1 or Nsl1 had been present in mESCs, the NSAF values would have been much higher for Msl1 than for Mof, or for Nsl1 than for Mof, which was clearly not the case.

In addition, as required we further addressed the reviewers’ concern and performed a gel filtration (GF) experiment using nuclear extracts prepared from mouse ESCs as described by the new Figure 1. This new data (shown in new Figure 1) now further indicates that Msl1 and Nsl1 incorporate in their respective complexes (together with Mof), which elute from the GF column at their respective molecular weights (250 kDa for MSL and 760 kDa for NSL). Moreover, we did not detect any free Msl1, or Nsl1 (in the 65 kDa and 130 kDa range), respectively) present in mESC nuclear extracts. Thus, we can assume that Msl1 and Nsl1 ChIP-seq profiles are representative for MSL and NSL complex binding (however see also answer to point 2). Moreover, our new GF results showing the elution of Mof in the 50 kDa range (without Nsl1 and Msl1), suggest the existence of a small “free“ Mof pool in ESCs.

We think that we have now convincingly demonstrated (as originally stated) that Msl1 or Nsl1 are exclusively present in their respective HAT complexes, in which Mof is present with similar abundance (with a close to 1:1 subunit ratio) than either Msl1, or Nsl1.

*4) The overlap between MSL / NSL and bivalent genes appears not very significant (*Figure 6
*– only cluster C contains bivalent genes). The large majority of Msl1 and Nsl1 bound genes are not H3K27me3. Throughout the manuscript genes are categorized but statistical testing seems to be not performed. Rigorous statistical tests and control groups need be included to establish meaningful correlations*.

The overlap between MSL and NSL with bivalent sites is now included in our revised manuscript in Figure 8. We focused on all MSL and NSL binding sites and analysed the overlap with H3K27me3, Ezh2 and H3K4me3, which are common markers for bivalent genes (Figure 8). Following the reviewers’ concerns we have conducted Bootstrap statistical analysis, which is explained in the Materials and methods section. Indeed, our statistical analyses demonstrated that the identified number of bivalent genes is significantly enriched compared to a random selection of genes (see new Figure 8—figure supplement 1). The analysis, the result and the obtained p-value are further described in the legend of Figure 8—figure supplement 1.

*5) Acetylation of other residues than H4K16ac is suggested (Discussion) as a redundant mechanism. It is not clear how to reconcile this speculation fits with the Mof phenotype*.

We agree with the reviewers that our speculation did not fit with the Mof phenotype. Therefore, as required we have now deleted the original Figure 6—figure supplement 1 and have re-written the corresponding paragraph “MSL is the main H4K16 HAT in mESCs” in the Discussion.